

# Unveiling the organic contribution to the initial particle growth in 3-10 nm size range

Kewei Zhang[1], Zhengning Xu[1,2], Fei Zhang[1,2], Zhibin Wang[1,2,*]

[1]State Key Laboratory of Soil Pollution Control and Safety, Zhejiang Provincial Key Laboratory of Organic Pollution Process and Control, College of Environmental and Resource Sciences, Zhejiang University, Hangzhou 310058, China
[2]ZJU-Hangzhou Global Scientific and Technological Innovation Center, Zhejiang University, Hangzhou 311200, China

*Correspondence to*: Zhibin Wang (wangzhibin@zju.edu.cn)

## Abstract

Organic compounds play an important role in driving atmospheric particle formation and initial growth along with sulfuric acid. However, the detailed chemical composition of newly formed particles remains limited due to analytical challenges. In this study, we conducted the laboratory experiments using a flow tube reactor to investigate the roles of sulfuric acid (SA) and oxygenated organic molecules (OOMs, from α-pinene oxidation) in nanoparticle growth. For the first time, the size-resolved hygroscopicity parameter ($\kappa$) and organic mass fraction ($f_{org}$) of 3-10 nm particles were measured using a custom-built scanning flow condensation particle counter (SFCPC). The hygroscopicity of SA decreased 49% as particle size increased (from 0.413 ± 0.011 at 3 nm to 0.209 ± 0.004 at 10 nm) and declined by up to 18% with increasing RH, primarily due to hydration effects. In contrast, the $\kappa$ values of OOMs increased with RH by as much as 57%, attributable to changes in oxidation product. Size-resolved $f_{org}$ revealed that larger particles contained a greater proportion of organics, indicating that SA dominates the growth of small particles, whereas OOMs contribute more significantly to growth at larger sizes. Moreover, elevated humidity enhanced the organic contribution to particle growth by up to 81%. This enhancement was more pronounced for 5-10 nm particles due to the incorporation of increased yields of more volatile oxidation products and Kelvin effect. These valuable information on hygroscopicity and chemical composition of 3-10 nm particles during new particle formation and subsequent growth could further the understanding of related atmospheric mechanisms.



# 1 Introduction

Atmospheric new particle formation (NPF) is a widely observed phenomenon (Du et al., 2024; Kulmala et al., 2014) in which low-volatility gas-phase oxidation products nucleate to form aerosol particles (Lee et al., 2019). These newly formed particles can grow to become cloud condensation nuclei (CCN), potentially contributing 30-70% of atmospheric CCN populations (Ren et al., 2021; Sun et al., 2024). The chemical composition of nucleating clusters and growing particles plays a fundamental role in determining both particle formation mechanisms and subsequent growth processes (Kirkby et al., 2023). However, the chemical information of the cluster and particles are still not well understood (Zhang et al., 2012; Zhao et al., 2024).

To date, the condensation of low volatility vapours, like sulfuric acid (SA) or oxidized organic compounds, is recognized as a primary mechanism driving cluster formation and particle growth (Stolzenburg et al., 2018, 2020). SA, in particular, is a key gas-phase precursor involved in the nucleation of atmospheric aerosol particles (Kulmala, 2003; Sipilä et al., 2010). Field observations (Kulmala et al., 2013; Wang et al., 2011; Yao et al., 2018; Zhang et al., 2009) and laboratory experiments (Dunne et al., 2016; Kirkby et al., 2023; Sipilä et al., 2010) demonstrate it also contributes to initial growth. But SA only contributes 10%-50% exceeding 3 nm (Stolzenburg et al., 2023), which indicates that SA alone rarely dominates nanoparticle growth in the atmosphere (Kuang et al., 2010). In addition, α-pinene oxidation products contribute to the growth of newly formed particles (Ehn et al., 2014; Tröstl et al., 2016). Atmospheric organic vapors play a crucial role in particle formation and growth (Kirkby et al., 2016; Riccobono et al., 2014; Zhang et al., 2004), with organic compounds potentially accounting for a substantial fraction (20-90%) of submicron particle mass (Jimenez et al., 2009). And the contribution of organic species to particle growth was found to increase with particle size (Bianchi et al., 2019; Riccobono et al., 2012; Riipinen et al., 2012). However, quantitative analysis of SA or organic mass fractions in 3-10 nm particles remains challenging due to instrumental limitations in measuring chemical composition (Smith et al., 2021; Zhang et al., 2022).

Several techniques have been developed to address this measurement gap. The thermal desorption chemical ionization mass spectrometer (TDCIMS) could characterize particle composition down to 8 nm in field (Li et al., 2021), though requiring extended sampling periods (10-30 minutes). Keskinen et al. (2013) estimated the organic fraction in sub-2 nm using atmospheric pressure interface time-of-flight (API-TOF) mass spectrometer. To break the limitation of insufficient ion concentration, alternative approaches utilizing advanced condensation particle counters (CPCs) provide qualitative assessments of organic contributions (Kangasluoma et al., 2014; Kulmala et al., 2007; O'Dowd et al., 2002), by exploiting the inherently high number concentration of nucleation-mode particles. The nano cloud condensation nuclei counter (nano-CCNC) could further provide semi-quantitative information by applying the linear relationship between hygroscopicity parameter ($\kappa$) and organic mass fraction ($f_{org}$) down to 2.5 nm (Wang et al., 2015). Though its stabilization period (~2 minutes for a single supersaturation) limits applicability to rapidly growing newly formed particles (e.g., 35.7 nm h$^{-1}$ in urban Shanghai, Xiao et al., 2015). More recently, the scanning flow condensation particle counter (SFCPC) has demonstrated capability for the $\kappa$-$f_{org}$





relationship by achieving supersaturation adjustments within ∼2 seconds through flow rate modulation (Zhang et al., 2023), making it particularly suitable for organic fraction determination in 3-10 nm.

In this study, we conducted a series of laboratory nucleation and growth experiments using a custom-built flow tube reactor. $SO_2$ and α-pinene were employed as gas-phase precursors to generate SA and organics (OOMs, oxygenated organic molecules), respectively. Experiments were performed under purely inorganic, purely organic, and mixed precursor conditions with varying [α-pinene]/[$SO_2$] ratios, across a wide RH range (20%-80%), to investigate the role of organics in the initial particle growth. We first measured the $\kappa$ values of 3-10 nm particles formed from the oxidation products through SFCPC. Then our analysis established size-resolved linear relationships between $\kappa$ and $f_{org}$ for SA-OOMs mixed particles, enabling

quantitative determination of organic contributions. Furthermore, we systematically examined the effects of particle size, gas precursor concentration ratio, and humidity on both particle hygroscopicity and organic contribution.

## 2 Materials and methods

### 2.1 Experimental set-up

A custom-built flow tube reactor was used to perform a series of laboratory studies on nucleation and growth. This flow

tube consisted of a 25 cm long × 1 cm i.d. quartz tube (19.6 mL in volume) fitted with stainless steel adapter on each end. The entrance was coupled with two union cross in line to introduce gas precursors. As shown in Figure 1, the water vapor generated by passing $N_2$ through water and $O_3$ from a UVP ozone generator (model SOG-2, Analytik Jena US) were introduced into the main gas flow in the first union cross. The SA and OOMs in all experiments were generated from $SO_2$ and α-pinene. These precurses were diluted from a gas cylinder containing 0.6% $SO_2$ and obtained from a gas cylinder with 50 ppmv α-pinene,

respectively. The carrier flow, water vapor, $O_3$, $SO_2$ and α-pinene were mixed in the second union cross and then introduced into flow tube reactor. To initiate photolysis reactions in the system, a UV lamp (model 11SC-1, Analytik Jena US) with a length of 5.38 cm was installed at the entrance of the quartz tube, emitting ultraviolet light at a wavelength of 254 nm. At the exist of flow tube reactor, the temperature and RH were measured by a humidity sensor (model SHT85, Sensirion) with precision of ± 1.5%. The $O_3$ concentration was monitored by an ozone analyser (model 49i, Thermo Fisher Scientific). The

concentrations of other gaseous precursors were derived from their mixing ratios, and the molecular composition of the flow tube products was not directly measured in this study.



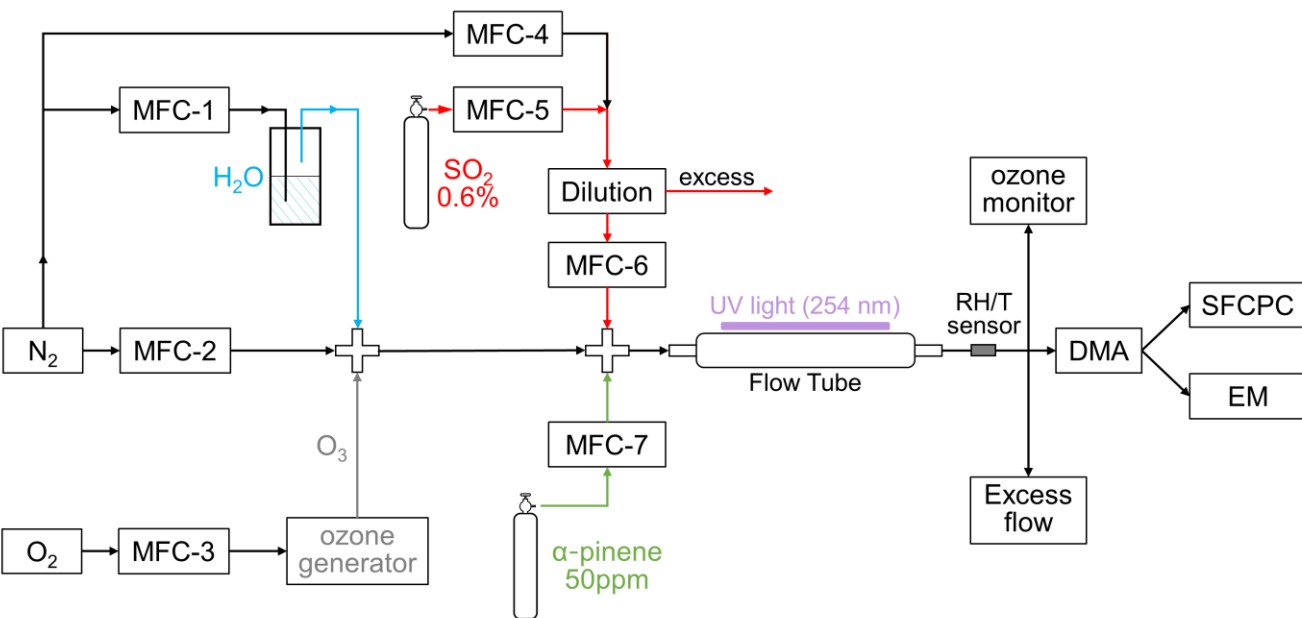

**Figure 1. Schematic of the flow tube reactor and experimental setup. Precursors (SO₂, α-pinene, O₃ and H₂O) were introduced into the flow tube at a total flow rate of ~3 L min⁻¹. SO₂ and α-pinene were subsequently oxidized by O₃ under UV irradiation (254 nm). The resulting oxidation products were classified using a nano-DMA, with particle hygroscopicity and chamical compostion characterized by SFCPC and EM.**

SA and OOMs was used in this work to represent inorganic and organic components in atmosphere respectively. For pure inorganic experimental groups, $H_2SO_4$ was generated in situ via the reaction of OH radicals with $SO_2$ in the presence of water vapor. When the $SO_2$, $O_3$ and $H_2O$ were mixed and introduced into the flow tube reactor, $O_3$ underwent photolysis to produce $O(^1D)$ atoms, which then react with water vapor to generate OH radicals. These OH radicals interact with $SO_2$, forming the $HOSO_2$ adduct, which subsequently decomposed to produce $SO_3$ and $HO_2$ (Jayne et al., 1997; Lovejoy et al., 1996). In the reaction of $SO_3$ with water vapor, two $H_2O$ molecules or one $H_2O$ dimer per $SO_3$ molecule were required, ultimately resulting in the formation of $H_2SO_4$ (Berndt et al., 2005). For pure organic experimental groups, OOMs were produced from organic peroxides formed from oxidation reactions of α-pinene (Kirkby et al., 2016; Lee et al., 2019). α-pinene was exposed to ozone and also to hydroxyl radicals (OH·) due to the unavoidable production of OH· from ozone photolysis and secondary reactions.

The $SO_2$ and α-pinene concentration in experiments were regulated by setting the mixing ratio of $SO_2$ flow rate and α-pinene flow rate to the total flow rate. The total flow rate was 3000 mL min⁻¹, and the residence time in flow tube reactor was about 0.4 s. The all flow rates in this flow tube reactor were set by mass flow controllers (MFCs; MFC. 1-4, model GT130D, Gas Tool Instruments Co., Ltd.; MFC. 5-7, model Sevenstar CS200, NAURA Technology Group Co., Ltd.). The conditions of 24 experimental groups conducted to research the organic contribution to new formed particles in the initial growth stage were summarized in Table S1. To research the RH impacts to the initial growth process, the experiments were divided to four series conducted under RH 20%, 40%, 60% and 80%. For pure inorganic (Exp. A) and mixture groups (Exp. C-F), the concentration of $SO_2$ was set as a constant value and the concentration of α-pinene was regulated based on the [α-pinene]/[SO₂]



(concentration ratio of gas precursors α-pinene and SO$_2$, 0.1-1). The pure organic experiments (Exp. B) were conducted with

much higher concentration than that in mixture experimental groups to generate sufficient 3-10 nm particles (number

concentration larger than 1000 # cm$^{-3}$). The temperature in all experiments was around 28 ℃ when the reaction was stable. O$_3$

concentration obtained by Ozone monitor was ~200 ppb with a regulable O$_2$ flow rate.

## 2.2 Determination of $\kappa$

The hygroscopicity of nanoparticles was measured with a custom-designed SFCPC. The setup of SFCPC system has been

described previously (Zhang et al., 2023), and only a brief summary is presented here. The aerosol particles were passed

through a neutralizer (X-ray, TSI model 3080), and a nano-differential mobility analyzer (nano-DMA, TSI model 3086) was

used to select charged monodisperse particles in dimeter range of 3-10 nm (with intervals of 0.2 nm in the 3-4 nm range and

intervals of 0.5 nm in the 4-10 nm range). SFCPC, improved from a water-based condensation particle counter (WCPC, TSI

model 3788), was deployed to change the supersaturation fast by altering the sample flow. And an electrometer (EM, TSI

model 3068B) was operated in parallel to obtain the counting efficiency of sampled particles ($f_{count}$) in different supersaturation

($S$) conditions. $S$ distributions of SFCPC were calibrated by tungsten oxides (WO$_x$) particles generated from a WO$_x$ generator

(Model 7.860, Grimm Aerosol Technik; Steiner, 2006), and twelve $S$ conditions were setup to meet the requirements of $\kappa$ value

measurements. The effective $S$ ranged from 7.7% to 73.1%, a sufficiently broad range to activate the SA and OOMs

corresponding $f_{count}$ in 20%-80%. For the measurement part, particles from flow tube reactor were selected with discrete

diameters and the $S$ could be calculated through the measured $f_{count}$. Then the $\kappa$ values could be obtained from dry diameter

($D_d$) and its corresponding $S$ based on $\kappa$-Köhler equation (Petters and Kreidenweis, 2007).

## 2.3 Determination of $f_{org}$

The linear relationship between chemical composition and hygroscopicity of CCN sized particles has been found both in

lab experiments and field measurements (Dusek et al., 2010; Pöhlker et al., 2023; Vogel et al., 2016; Zhou et al., 2024). Our

previous work extended the linear relationship between $\kappa$ and $f_{org}$ to 3-10 nm for AS and levoglucosan/sucrose mixed particles

(Wang et al., 2015; Zhang et al., 2023). In this work, we established $\kappa$-$f_{org}$ linear relationship in SA-OOMs mixture system

according to the measured $\kappa_{org}$ and $\kappa_{inorg}$ in pure organic and inorganic experiment groups. Then the $f_{org}$ of particle products

from flow tube reactor could be obtained by Eq. 1:

$$f_{org} = \frac{\kappa - \kappa_{inorg}}{\kappa_{org} - \kappa_{inorg}} \times 100\% \tag{1}$$

where $f_{org}$ and $\kappa_{org}$ is the mass fraction and hygroscopicity of pure organics (OOMs in this work), respectively; $\kappa_{inorg}$ is the

hygroscopicity of pure inorganic component (SA in this work); $\kappa$ is the hygroscopicity of the measured particles nucleation

and growth process in flow tube reactor.



## 3 Results and discussion

### 3.1 Hygroscopicity

The results of pure inorganic groups and organic groups were shown in Figure 2. The hygroscopicity of 3-10 nm SA particles formed by the oxidation products of $SO_2$ have significant size dependence. $\kappa$ values would decrease with $D_d$, from $0.413 \pm 0.011$ for 3 nm to $0.209 \pm 0.004$ for 10 nm under RH = 20%. It should be noted that the quartz tube was replaced for every single experiment group to eliminate the potential contamination between experiments. Moreover, the mass spectrometry did not detect any obvious organic compound signals, so this decrease trend did not caused by organic contamination.

Considering the sulfuric acid and water binary nucleation is the basic mechanism for the SA particle formation, the water molecules plays a significant role through hydration (Kulmala et al., 1998; Lee et al., 2019; Stolzenburg et al., 2023; Yu et al., 2017). The initial formation of $H_2SO_4 \cdot H_2O$ molecular clusters is followed by rapid addition of further $H_2O$ molecules and these stepwise hydrates process ultimately leads to particles covered with water during growth (Couling et al., 2003; Matsubara et al., 2009). The measured particles were thought as $(H_2SO_4)_m\text{-}(H_2O)_n$ rather than pure $H_2SO_4$. Consequently, the decreasing

trend of $\kappa$ with particle size may be explained by the increasing water content, which lowers the particles' water uptake capacity relative to their dry mass.

Hygroscopicity of SA particles also revealed a consistent decrease with rising RH: at RH = 80%, $\kappa$ values declined to $0.361 \pm 0.013$ for 3 nm and $0.171 \pm 0.011$ for 10 nm particles, decreasing 13% and 18% compared to RH = 20%. This demonstrates that SA particles exhibit reduced hygroscopicity under higher humidity conditions. Previous studies have shown

that the average number of water molecules hydrating each $H_2SO_4$ molecule increases with RH (Kurtén et al., 2007; Temelso et al., 2012; Zollner et al., 2012). Therefore, the observed decrease in hygroscopicity can be attributed to the particles already containing more water at higher RH, resulting in a reduced capacity for additional water uptake.

The $\kappa$ values reported in previous studies are summarized in Table 1. The measured hygroscopicity results in this work are much lower than those reported in previous studies (0.68-0.9, Petters and Kreidenweis, 2007; Shantz et al., 2008; Sullivan

et al., 2010), which predicted by thermodynamic model in 30-80 nm based on the model parameters provided by Clegg et al. (1998). The hygroscopicity of newly formed sulfuric acid nanoparticles in CLOUD chamber was examined by a nano hygroscopicity tandem differential mobility analyser (nano-HTDMA) and the reported $\kappa$ values were $0.64 \pm 0.02$ and $0.52 \pm 0.02$ for 10 nm and 15 nm, respectively (Kim et al., 2016). The reported variation trend in particle size aligns with our findings but the $\kappa$ values were much larger than that in this work. The measurement condition would induce inherent different $\kappa$ values,

where SA particles are expected to exhibit comparably higher hygroscopic growth at subsaturated conditions and lower CCN activity at supersaturation (Biskos et al., 2009; Massling et al., 2023). In addition, the nano size effect on the thermodynamic properties of aerosol particles is also a significant contributing factor (Cheng et al., 2015). As far as we know, direct measurements of the $\kappa$ of sulfuric acid particles remain limited in the literature, and our results is the first measurement under supersaturation condition in 3-10 nm range.





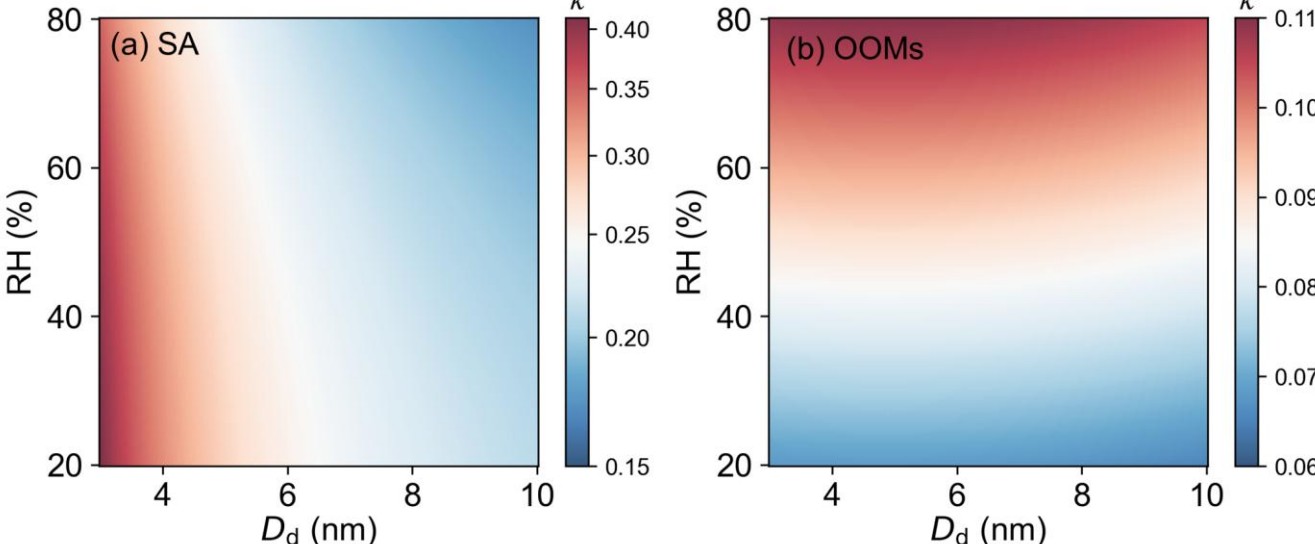


**Figure 2. Dependence of hygroscopicity on particle size and RH: (a) SA formed from SO₂, and (b) OOMs from α-pinene oxidation. Color scale represents the $\kappa$ values derived from $f_{org}$ measurements using SFCPC and EM, where the values were fitted from measurement data to illustrate $\kappa$ variation trend more clearly.**

Differ from the SA results, the OOMs particles formed by the oxidation products of α-pinene have almost constant $\kappa$

values in 3-10 nm diameter range. As shown in Figure 2(b), $\kappa$ of the pure organic group under 20% RH is ~0.065. This results indicated that the OOMs particles exhibit non-hygroscopicity compared to SA particles and its hygroscopicity has no significant dependence on particle diameter. Alfarra et al. (2013) measused hygroscopicity of particles formed from α-pinene oxidation via both OH radicals and O₃ using CCNC. Our results align closely with their reported $\kappa$ values (0.1-0.16), which were also observed under supersaturation conditions. Furthermore, numerous studies have investigated the hygroscopicity of

α-pinene oxidation products using CCNC or HTDMA, reporting a wide range of $\kappa$ values (0.03-0.19, Cain et al., 2021; Duplissy et al., 2011; Engelhart et al., 2008; Massoli et al., 2010; Razafindrambinina et al., 2022; Wang et al., 2019; Zhao et al., 2016). All experimental results reported here fall within the established range of values found in existing publications. However, to our knowledge, no studies have examined the particle size dependence of hygroscopicity for particles from α-pinene oxidation, particularly in the 3-10 nm size range.

Contrasting with the negligible size dependence, the measured $\kappa$ values of OOMs exhibited a pronounced increase with RH, rising by 57% from 20% to 80% RH. This finding aligns with Razafindrambinina et al. (2022), who reported higher $\kappa$ values for laboratory-generated α-pinene oxidation products under humid conditions ($\kappa$ = 0.191 at 75-80% RH) compared to dry conditions ($\kappa$ = 0.130 at RH < 10%). Similarly, Luo et al. (2024) observed that the molecular composition of α-pinene oxidation products evolves with increasing RH. Moreover, the work of Yuan et al. (2017) suggests that in the presence of

water vapor, particles formation may promote the generation of more stable Criegee intermediates, leading to the formation of more hygroscopic materials in monoterpene systems. The observed increase in $\kappa$ values at higher RH in this study may thus





be attributed to the production of components with stronger hygroscopicity, likely corresponding to the presence of more oxygenated functional groups (Han et al., 2022), such as multifunctional carboxylic acids (Poulain et al., 2010).

**Table 1. Summary of $\kappa$ values of SA and oxidation products of $\alpha$-pinene reported in previous studies and this work.**

|  | $\kappa$ values | Method | $D_d$ (nm) | RH | Reference |
|---|---|---|---|---|---|
|  | 0.9 | model | - | - | Petters and Kreidenweis (2007) |
|  | 0.68-0.74 | model | 30-80 | - | Shantz et al. (2008) |
|  | 0.7 | model | - | - | Sullivan et al. (2010) |
|  | 0.64 ± 0.02 | HTDMA | 10 | 38% | Kim et al. (2016) |
|  | 0.52 ± 0.02 |  | 15 |  |  |
| SA | 0.413 ± 0.011 |  | 3 | 20% |  |
|  | 0.209 ± 0.004 |  | 10 |  |  |
|  | 0.400 ± 0.019 |  | 3 | 40% |  |
|  | 0.199 ± 0.008 | SFCPC | 10 |  | This work |
|  | 0.379 ± 0.017 |  | 3 | 60% |  |
|  | 0.194 ± 0.004 |  | 10 |  |  |
|  | 0.361 ± 0.013 |  | 3 | 80% |  |
|  | 0.171 ± 0.011 |  | 10 |  |  |
|  | 0.1-0.16 | CCNC | 20-500 | 51.4%-71.4% | Alfarra et al. (2013) |
|  | 0.04-0.12 | HTDMA | - |  |  |
|  | 0.11 | CCNC | - | - | Cain et al. (2021) |
|  | 0.11-0.19 | CCNC | 10-450 | 60%-70% | Zhao et al. (2016) |
|  | 0.03-0.06 | HTDMA | - |  |  |
|  | 0.130 ± 0.019 | CCNC | 8-352 | <10% |  |
|  | 0.059 ± 0.019 | HTDMA | 200, 250, 300 |  | Razafindrambinina et al. (2022) |
|  | 0.191 ± 0.013 | CCNC | 8-352 | 61% |  |
| OOMs | 0.042 ± 0.013 | HTDMA | 200, 250, 300 |  |  |
|  | 0.065 ± 0.004 |  | 3 | 20% |  |
|  | 0.066 ± 0.010 |  | 10 |  |  |
|  | 0.076 ± 0.007 |  | 3 | 40% |  |
|  | 0.076 ± 0.002 | SFCPC | 10 |  | This work |
|  | 0.100 ± 0.006 |  | 3 | 60% |  |
|  | 0.101 ± 0.009 |  | 10 |  |  |
|  | 0.104 ± 0.006 |  | 3 | 80% |  |
|  | 0.102 ± 0.002 |  | 10 |  |  |


## 3.2 Organic mass fraction

The $f_{org}$ values of 3-10 nm particles was determined using the $\kappa$-$f_{org}$ linear relationship as described in Section 2.3. Given the pronounced size dependence of $\kappa$ for SA particles within the 3-10 nm range, we developed size-resolved linear relationships.



To minimize measurement uncertainties, the values of $\kappa_{inorg}$ and $\kappa_{org}$ used in size-resolved $\kappa$-$f_{org}$ linear relationship were derived
from fitted trends between $\kappa$ and $D_d$ in SA and OOMs experiment groups. Then the OOMs mass fraction was subsequently
calculated using Equation 3, based on the measured $\kappa$ values of SA-OOMs mixture particles.

Figure 3 presents the retrieved $f_{org}$ results for four experimental groups with varying [α-pinene]/[$SO_2$] ratios under 20%
RH. For 3-10 nm SA-OOMs mixed particles, the median mass fraction of OOMs increased from 7.94% to 36.86% as the [α-
pinene]/[$SO_2$] ratio increased from 0.1 to 1. Considering that ozone was always excessive in the flow tube reactor, the oxidation
products of α-pinene are expected to increase with rising precursor concentration. Consequently, the organic content in the
particle phase should be proportional to the concentration of condensable OOMs in the gas phase. This significant increasing
trend has also been reported in previous studies. Li et al. (2022) observed that the ratio of particulate organics to sulphate in
urban field measurements was positively correlated with the ratio of gaseous condensable organic oxidation products to sulfuric
acid. Comprehensive modelling study demonstrated that terpene-rich air masses containing abundant low volatility oxidation
products substantially enhanced the condensational growth of nano-particles and dominated the initial growth stage in sub-10
nm, with contribution as high as 95% (Huang et al., 2016). Our experimental results align with these findings, demonstrating
that increased organic precursor concentrations significantly elevates the contribution of OOMs to the growth of 3-10 nm
particles.

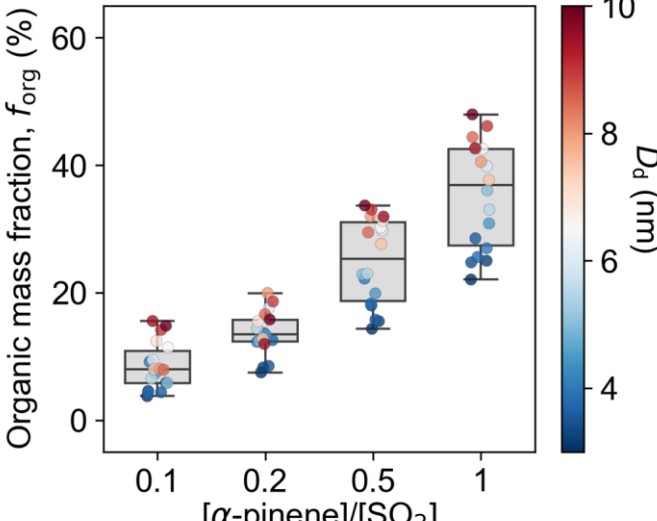

**Figure 3. Organic mass fraction as a function of the precursor concentration ratio for SA-OOMs mixture particles at RH 20%. The
box plots display the interquartile range (IQR), with the central line denoting the median and whiskers extending to 1.5×IQR. The
coloured scatter points represent the mean $f_{org}$ values for each selected $D_d$, as indicated by the colour scale.**

At the same time, our results reveal that the organic content of large particles is generally higher than that of smaller
particles (coloured scatter points in Figure 3). To further investigate the size-dependent effect on the chemical composition of
nano-particles, the experimental data were analysed by categorizing particles into three distinct size ranges (3-5 nm, 5-7 nm,
and 7-10 nm). Figure 4 shows the results from Exp. C and Exp. F, which had the lowest and highest [α-pinene]/[$SO_2$] ratios of



0.1 and 1, respectively. The $f_{org}$ values for 3-5 nm, 5-7 nm, and 7-10 nm SA-OOMs particles in Exp. C are 5.67%, 9.82% and 11.25%, respectively. Similarly, particles formed in Exp. F contained 27.47%, 39.49%, and 43.22% organics, respectively. These results indicate that the $f_{org}$ exhibited a consistent increase with $D_d$ across the range of the precursors mixing ratios examined in this work. Furthermore, these findings demonstrate that α-pinene oxidation products contributed progressively more to particulate phase in the SA-OOMs system as particle size increased.

Several studies have also investigated the chemical composition of newly formed particles in the initial growth stage. Kim et al. (2016) employed nano-HTDMA measurements in the CLOUD chamber to determine $\kappa$ values, from which volume fractions were derived through linear relationships. Their work revealed that the volume fraction of dimethylamine sulphate in sulfuric acid-dimethylamine systems increased substantially from 0.20-0.29 for 10 nm particles to 0.58-0.92 at 15 nm. Similarly, Keskinen et al. (2013) observed a progressive enhancement in organic volume fraction from 0.4 at 2 nm to 0.9 at 63 nm, while noting the existing measurement challenges for particles between 2-15 nm where chemical characterization remains particularly difficult. More recently, Li et al. (2022) achieved direct measurements of size-resolved molecular composition using TDCIMS, demonstrating a clear increase in organic mass fraction with particle diameter across the 8-40 nm range in urban Beijing. Other studies have also indicated that particle growth mechanisms exhibit a dependence on particle size (Riipinen et al., 2012). The contribution of SA decreased as particle size increased (Xiao et al., 2015), while organic compounds increasingly dominated the growth process, showing a strong size-dependent effect (Riccobono et al., 2012). Our experimental results align well with these previous studies, confirming that organic contributions to nanoparticle growth exhibit consistent size dependence, while additionally providing novel composition data for freshly nucleated particles as small as 3 nm, thereby extending the current understanding of early particle growth.

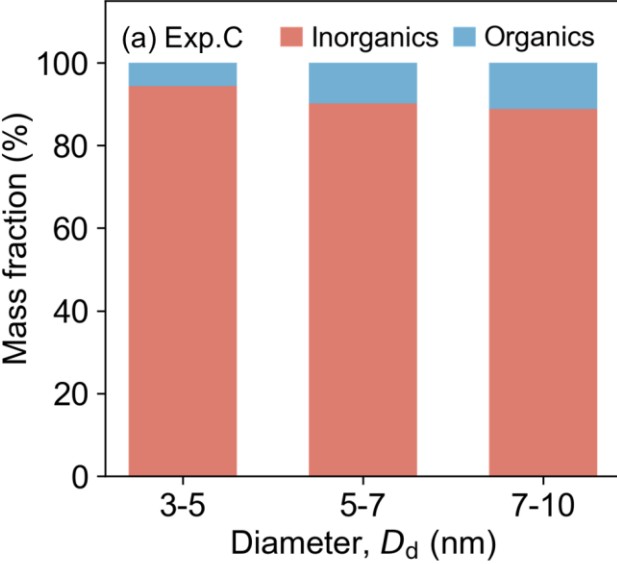
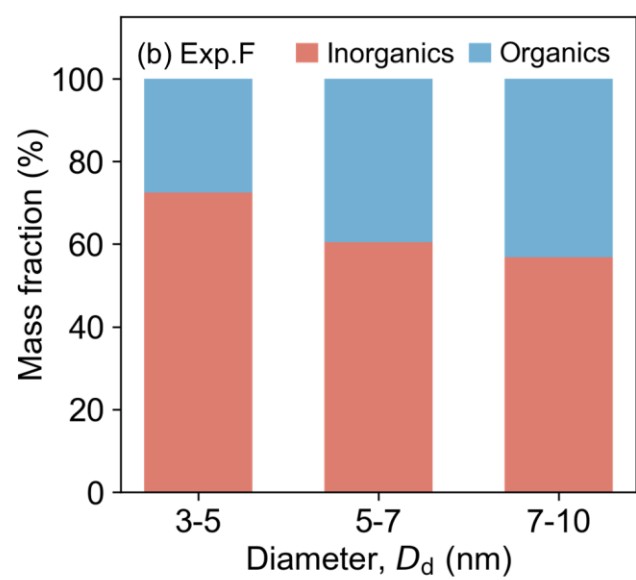





**Figure 4. Organic mass fraction of SA-OOMs mixtures as a function of particles diameter range with two precursors mixing ratios: (a) 0.1 for Exp. C, (b) 1 for Exp. F. Red and blue bars represent $f_{org}$ values of inorganics (SA) and organics (OOMs). Note that the size ranges are defined as semi-open intervals, except for the first bin which includes its lower bound.**

### 3.3 Effect of relative humidity

To investigate the effect of humidity on the initial growth of newly formed particles, we increased RH from 20% to 80% and measured the organic content across this range. The retrieved $f_{org}$ values were similarly grouped by particle size, following the same methodology as in the size-dependence analysis. The results from all RH experimental groups (20%-80%) were compared with those under 20% RH to examine the relative change in organic contribution with increasing RH (Figure 5). The enhancement of organic contribution to particle growth by humidity exhibits a marked difference between particles below and above 5 nm. Taking Exp. D as an example, the $f_{org}$ for 3-5 nm particles began to increase from 60% RH, with a relative increment reaching approximately 47% at 80% RH. For particles in the 5-7 nm and 7-10 nm size ranges, $f_{org}$ exhibited a sharp increase starting at 40% RH and remained almost stable within the 40%-80% RH range. Compared to the condition at 20% RH, the organic contribution to particle growth was enhanced by about 52% and 81%, respectively. This behaviour is attributed to changes in the properties of oxidation products under different humidity conditions. For such small nanoparticles, the partitioning of a molecule into the particulate phase is influenced by both its volatility and the Kelvin effect (Riipinen et al., 2012). Previous molecular measurements in both gas and particle phases have reported increased yields at elevated RH (Emanuelsson et al., 2013; Poulain et al., 2010). Concurrently, Surdu et al. (2023) observed that α-pinene oxidation products become more volatile under humid conditions. These competing mechanisms may explain the relative stability of $f_{org}$ in smaller particles. For larger particles, the diminished Kelvin effect facilitates the condensation of organic compounds, allowing even more volatile products to contribute to nanoparticle growth. Overall, increased RH enhances the organic contribution by altering the properties of α-pinene oxidation products, with a more pronounced effect observed for larger particles.




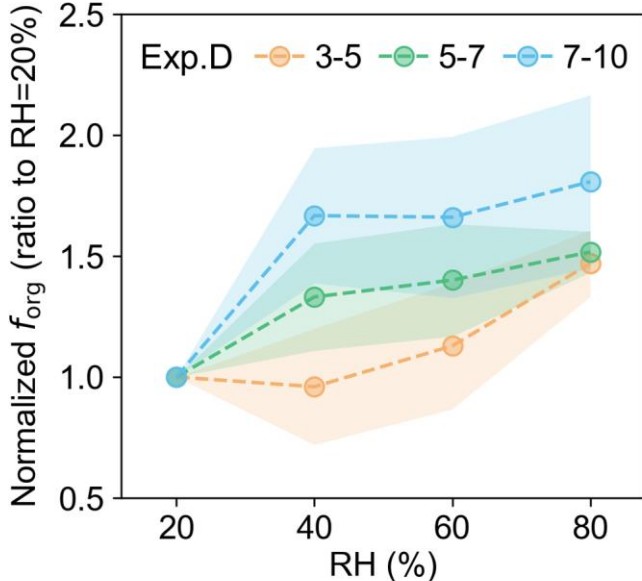

**Figure 5. Relative change in $f_{org}$ for 3-5 nm, 5-7 nm, and 7-10 nm particles as a function of RH. Data points indicate $f_{org}$ values**
**normalized to measurements at 20% RH. Shaded areas represent the standard deviation for each size range.**

## 4 Conclusion

In this study, a flow tube reactor was employed to conduct a series of experiments investigating nucleation and particle growth across a wide range of RH (20%-80%). $SO_2$ and α-pinene served as gas-phase precursors for the formation of SA and OOMs in all experiments, with the precursor concentration ratio [α-pinene]/[$SO_2$] varying from 0.1 to 1. A custom-designed
SFCPC was utilized to quantify the size-resolved hygroscopicity. Based on the linear relationship between $κ$ and $f_{org}$, the organic mass fraction of freshly formed particles was determined within the 3-10 nm diameter range.

Our work provides quantitative insight into the distinct hygroscopic behaviours of SA and OOMs. For SA, $κ$ exhibited significant size dependence, decreasing from $0.413 ± 0.011$ at 3 nm to $0.209 ± 0.004$ at 10 nm under 20% RH. When RH was elevated to 80%, $κ$ further declined by 18% (to $0.171 ± 0.011$ at 10 nm). This decrease is interpreted as the formation of
$(H_2SO_4)_m$-$(H_2O)_n$ complexes. The hydration process eventually results in water-covered particles, reducing their capacity for further water uptake. The measured $κ$ values in this work are much lower than those reported in previous studies and this discrepancy may be attributed to the nano size effect and different measurement techniques. In contrast, $κ$ of OOMs remained relatively constant across the 3-10 nm size range but increased with rising RH (from $0.069 ± 0.003$ to $0.108 ± 0.004$ as RH rose from 20% to 80%) due to changes in the physicochemical properties of the oxidation products. These experimental
findings are consistent with previously reported $κ$ values for α-pinene oxidation particles, which typically range from 0.03 to 0.19. The derived organic content of SA-OOMs mixed particles consistently increased with higher [α-pinene]/[$SO_2$] ratios, indicating enhanced contribution of organics to the particulate phase. Furthermore, $f_{org}$ exhibited a consistent increase with



particle size. Specifically, $f_{org}$ values increased from 5.67% to 11.25% and from 27.47% to 43.22% for [α-pinene]/[SO$_2$] of 0.1 and 1, respectively. This demonstrated that the contribution of α-pinene oxidation products to particulate phase in the SA-

OOMs system became more pronounced at larger particles. Our results align with previous studies, further confirming the size dependence of organic contributions to nanoparticle growth and additionally providing novel composition data down to 3 nm. The effects of RH on chemical composition revealed distinct trends in $f_{org}$ across different size ranges: for 3-5 nm particles, $f_{org}$ varied negligibly until 60% RH, while for 5-10 nm particles, it increased with RH and had nearly stabilized at 40% RH. These observations can be attributed to a combination of factors including the Kelvin effect as well as increased volatility and higher

yields under elevated RH.

To the best of our knowledge, this is the first study to measure the hygroscopicity of SA and OOMs particles down to 3 nm. Given that direct measurements of chemical composition for newly formed 3-10 nm particles remain limited in the literature, our study provides important, quantitative, and size-resolved organic content data in this nano size range. The experimental results of SA-OOMs mixture indicate that OOMs contribute significantly to the particulate phase, with their mass

fraction increasing with particle size, [α-pinene]/[SO$_2$] ratio, and RH. These findings provide valuable supplementary information for advancing our understanding of new particle formation and subsequent growth. Looking ahead, exploring of multi-precursor systems, longer oxidation times for better simulating aging processes, and the further deployment of the SFCPC in field measurements will yield deeper insights into the chemical composition of atmospheric aerosols.



**Author contributions:** KZ and ZW contributed to the methodology, data curation, and writing of the original draft. ZX and FZ contributed to the reviewing and editing. ZW contributed to the supervision, funding acquisition, conceptualization, investigation, data curation, writing, reviewing, and editing.

**Competing interests:** At least one of the (co-)authors is a member of the editorial board of Atmospheric Chemistry and Physics.

**Data availability:** Data are available upon request to the corresponding author.

**Acknowledgements:** This study was supported by the National Natural Science Foundation of China (41805100, 91844301, 42005086) and the National Key Research and Development Program of China (2022YFC3703505).



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
