# Peer review of "Unveiling the organic contribution to the initial particle growth in 3-10 nm size range"

_EGUsphere, 2025_

## Editor Comment (EC2)

*Comments:*

The authors investigate the influence mechanisms of sulfuric acid and oxygenated organic molecules on particle growth under varying particle sizes and RH conditions using a custom-built scanning flow condensation particle counter (SFCPC) to measure size-resolved hygroscopicity parameters and organic mass fractions in 3–10 nm particles. It further elucidates how the contribution of organic components evolves with particle size, particularly highlighting that increased RH significantly enhances the role of organics in the growth of 5–10 nm particles. This finding holds important value for understanding atmospheric new particle formation and growth mechanisms.

1. The authors treat the a-pinene and SO2 as the gas precursors, however, the true components participating particle nucleation should be OOM and H2SO4, which I guess that you did not measure in the current experiment setup. But still, I am wondering that does the [a-pinene]/[SO2] ratio could represent the ambient atmospheric condition?

2. Table 1 and Line 185: The authors claimed that the observed increase in $\kappa$ values at higher RH may be attributed to the production of components with stronger hygroscopicity, it there any direct evidence? What is the product in different RH condition and how to explain their influence on hygroscopicity?

3. Figure 5: It seems that the 3-5 nm particles are more affected by RH compared with larger size, why?

---

## Author Comment (AC1)

**Response to Editor**

**1. Comments and suggestions:**

*Please explain the novelty of this work in relation to your own previous work:*

*Zhang, K., Xiong, C., Cheng, Y., Ma, N., Mikhailov, E., Pöschl, U., et al. (2025). Assessment of hygroscopicity uncertainties associated with size and thermodynamic model: Implications for inferring chemical composition of sub‐10 nm particles. Journal of Geophysical Research: Atmospheres, 130, e2025JD043835. https://doi.org/10.1029/2025JD043835.*

**Responses and Revisions:**

We thank the editor for the constructive suggestions and comments on our manuscript entitled "Unveiling the organic contribution to the initial particle growth in 3-10 nm size range" [egusphere-2025-4421]. The novelty of this work and its differences from our previous study are summarized as follows:

Our previous work (Zhang et al., 2025) focused on the modeling studies of exemplary substances to assess uncertainties arising from the dependence of $\kappa$ on particle size and thermodynamic models. Although it discussed chemical composition inference (Section 3.4), its primary focus was on verifying the linear relationship of $\kappa$-$f_{org}$ and assessing the associated uncertainties. More critically, that work utilized particles generated via Electrospray from solutions of ammonium sulfate and *cis*-pinonic acid. This experimental approach, particularly the use of Electrospray to generate particles from non-volatile solutes, could not directly represent the gas-phase precursor mechanisms that govern atmospheric nucleation and growth processes.

The present study, by contrast, is designed to directly probe NPF and subsequent growth by simulating atmospheric processes. Utilizing a flow tube reactor, we introduced gas-phase precursors ($SO_2$ and α-pinene) to study nucleation and particle growth under controlled conditions. This methodology enables us to demonstrate the role of their oxidation products in the growth process. Thus, our work provides valuable and novel physical and chemical results of nanoparticle down to 3 nm that have not been previously reported, thereby yielding critical implications for understanding atmospheric particle formation and growth.

**2. Comments and suggestions:**

*In view of your own previous work cited above, consider rephrasing the following statements.*

*Line 14 in the abstract: "For the first time, the size-resolved hygroscopicity parameter*

*(κ) and organic mass fraction (ƒorg) of 3-10 nm particles were measured using a custom-built scanning flow condensation particle counter (SFCPC)."*

*Line 286 in the conclusion: "To the best of our knowledge, this is the first study to measure the hygroscopicity of SA and OOMs particles down to 3 nm."*

**Responses and Revisions:**

We thank the Editor for raising this point. We acknowledge that the original phrasing could be perceived as a broad overstatement by claiming the first-ever measurements of hygroscopicity ($\kappa$) and the organic mass fraction ($f_{\text{org}}$) in the 3-10 nm range in general. Our intention was to highlight that, to the best of our knowledge, this work provides the first measurements of size-resolved both the $\kappa$ and the $f_{\text{org}}$ results for nanoparticles in the 3-10 nm range formed from a sulfuric acid (SA) and α-pinene-derived oxygenated organic molecules (OOMs) system. The "first time" statements specifically refer to the SA-OOMs system, which was used in this work to simulate atmospheric new particle formation and subsequent growth processes.

To accurately reflect this specific contribution, we have revised the relevant statements in the manuscript:

(1) In the Abstract: "In this study, we conducted the laboratory experiments using a flow tube reactor to investigate the nanoparticle growth processes of sulfuric acid (SA) and oxygenated organic molecules (OOMs, from α-pinene oxidation) system. Utilizing a custom-built scanning flow condensation particle counter (SFCPC), we report, for the first time, size-resolved measurements of the hygroscopicity parameter ($\kappa$) and organic mass fraction ($f_{\text{org}}$) for particles in the 3-10 nm size range within this atmospherically relevant system."

(2) In the Conclusion: "To the best of our knowledge, this is the first study to measure the hygroscopicity ($\kappa$) of particles composed of SA and α-pinene-derived OOMs down to 3 nm."

**Refences:**

Zhang, K., Xiong, C., Cheng, Y., Ma, N., Mikhailov, E., Pöschl, U., Su, H., and Wang, Z.: Assessment of Hygroscopicity Uncertainties Associated With Size and Thermodynamic Model: Implications for Inferring Chemical Composition of Sub-10 nm Particles, J. Geophys. Res.-Atmos., 130, e2025JD043835, https://doi.org/10.1029/2025JD043835, 2025.

---

## Author Comment (AC2)

**Response to Referee #1**

**General Comments**
*This manuscript investigates the contribution of sulfuric acid and organic vapours (essentially alpha-pinene oxidation products) on the growth of sub-10 nm size particles as a function of particle size and relative humidity based on laboratory experiments. The investigation appears to be robust and novel enough to be published in a scientific publication. There are a few issues, however, that should be addressed before final acceptance of this work.*

**Responses and Revisions:**

We thank the reviewer for the constructive suggestions and comments concerning our manuscript entitled "Unveiling the organic contribution to the initial particle growth in 3-10 nm size range" [egusphere-2025-4421]. We sincerely appreciate these insightful comments, which have significantly improved both the quality of our manuscript and the methodological rigor of our ongoing research.

We have prepared point-by-point responses to the reviewer's comments below. The original reviewer comments appear in italics. Our responses are presented in blue plain font, while the corresponding revisions in the manuscript appear in blue underlined font.

**Scientific issues**

**1. Comments and suggestions:**
*Line 17-18: Hydration effects and changes in oxidation products are given as scientific facts for explaining the observations in the abstract. However, when reading the paper, it appears that these explanations, while likely, are speculations rather than facts. I recommend that the authors pay more attention what is interpreted as a scientific fact, a probable explanation, or speculation when discussing the observations.*

**Responses and Revisions:**

We sincerely thank the reviewer for this insightful and constructive comment. The reviewer is correct in pointing out that we should more carefully distinguish between established scientific facts and probable explanations based on our data and literature evidence. We agree that the terms "primarily due to" and "attributable to" in the original manuscript conveyed a degree of certainty that is not fully supported by our direct measurements. We have modified the wording in manuscript to clearly indicate that the proposed mechanisms are based on existing literature and are presented as plausible explanations rather than established facts.

The relevant text in the Abstract has been revised as follows:

"The hygroscopicity of SA decreased 49% as particle size increased (from $0.413 \pm 0.011$ at 3 nm to $0.209 \pm 0.004$ at 10 nm) and declined by up to 18% with increasing RH, which may be explained by hydration effects. In contrast, the $\kappa$ values of OOMs increased with RH by as much as 57%, potentially involving changes in oxidation product."

**2. Comments and suggestions:**
*Lines 59-66: This paragraph lists what will be done/investigated in the paper. The authors should also give either concrete scientific goals of the study or, alternatively, research questions aimed to be answered here.*

**Responses and Revisions:**

We sincerely appreciate the valuable suggestions and comments. As suggested, we have revised the relevant text in our manuscript (Section Introduction):

"In this study, we conducted a series of laboratory nucleation and growth experiments using a custom-built flow tube reactor. $SO_2$ and α-pinene were employed as gas-phase precursors to generate SA and organics (OOMs, oxygenated organic molecules), respectively. Experiments were performed under purely inorganic, purely organic, and mixed precursor conditions with varying [α-pinene]/[$SO_2$] ratios, across a wide RH range (20%-80%). We first measured the $\kappa$ values of 3-10 nm particles formed from the oxidation products through SFCPC. Then our analysis established size-resolved linear relationships between $\kappa$ and $f_{org}$ for SA-OOMs mixed particles, enabling quantitative determination of organic contributions. Furthermore, we systematically examined the effects of particle size, gas precursor concentration ratio, and humidity on both particle hygroscopicity and organic contribution. Based on these experimental results, this study aims to quantitatively investigate the distinct roles of sulfuric acid and oxygenated organics during nanoparticle growth, and to clarify how environmental conditions modulate the chemical composition and water uptake of sub-10 nm particles."

**3. Comments and suggestions:**
*Lines 249-250: This statement (This behaviour is attributed to ...) is given without a proper justification. The authors should add some reasoning(s), or at least speculations.*

**Responses and Revisions:**

We sincerely thank you for this insightful comment. You are right to point out that the statement regarding the attribution of the observed behavior needed further justification. In response to your suggestion, we have revised the manuscript to frame this explanation more appropriately as a plausible mechanism or speculation, which is

then supported by the subsequent reasoning and references. We have modified the relevant paragraph in the revised manuscript (Section 3.3):

"We speculate that these behaviours arise from the competing influences of humidity on the physicochemical properties of α-pinene oxidation products and the Kelvin effect. For such small nanoparticles, the partitioning of a molecule into the particulate phase is influenced by both its volatility and the Kelvin effect (Riipinen et al., 2012). Previous molecular measurements in both gas and particle phases have reported increased yields at elevated RH (Poulain et al., 2010). Concurrently, Surdu et al. (2023) observed that α-pinene oxidation products become more volatile under humid conditions. The relative stability of $f_{org}$ in the 3-5 nm particles at low RH condition may thus be explained by a balance between these two competing mechanisms, where the heightened Kelvin effect presents a significant barrier to condensation. For larger particles, the diminished Kelvin effect facilitates the condensation of organic compounds, allowing even more volatile products to contribute to nanoparticle growth. The distinct response patterns, where the enhancement occurred gradually for 3-5 nm particles but sharply for larger particles, suggest that the Kelvin effect plays a more dominant role for the smallest particle growth at lower RH. Overall, increased RH enhances the organic contribution by altering the properties of α-pinene oxidation products, with a more pronounced effect observed for larger particles."

**Technical issues**

**4. Comments and suggestions:**

*Lines 126-129 and 194-196: Although the procedure of determining the OOM mass fraction is relatively straightforward, the paper would benefit from having an example plot on how this works in practice for the data applied here.*

**Responses and Revisions:**

We thank the reviewer for this constructive suggestion. As suggested, we have now included an example plot in the Supplementary Information (Figure S1) to explicitly illustrate the practical procedure of how to establish size-resolved linear relationship between the hygroscopicity parameter ($\kappa$) and the organic mass fraction ($f_{org}$), and how the $f_{org}$ is retrieved from the measured $\kappa$. This figure demonstrates the application of our method in this study, thereby enhancing the clarity of our analytical approach.

Correspondingly, we have revised the relevant descriptions of method in the manuscript:

"In this work, we introduced $\kappa$-$f_{org}$ linear relationship into the mixing products from flow tube—a setup designed to simulate atmospheric processes—in order to

explore the organic content in the particulate phase. Furthermore, we established size-resolved $\kappa$-$f_{org}$ linear relationship to eliminate the mentioned uncertainty. The application of the $\kappa$-$f_{org}$ linear relationship relies on the assumption of ideal internal mixing within the particles. Under our experimental conditions, for in situ freshly formed 3-10 nm particles, the characteristic mixing times are short, and organic-inorganic mixtures are likely to remain liquid and well-mixed (Cheng et al., 2015). Therefore, the ideal internal mixing assumption is reasonable. For SA-OOMs mixture, we use the $\kappa$ values of pure organic (OOMs) and inorganic (SA) experiment groups to represent the hygroscopicity of the organic and inorganic component in the mixture particles, respectively. Although the organic and inorganic components in mixture may not be identical to those in pure organic and inorganic particles of the same size under the same RH due to the potential change of oxidation processes, this simplification is a necessary given the current inability to directly measure the composition and component-specific hygroscopicity of particles in the 3-10 nm size range. To further reduce uncertainties associated with the linear relationship, the $\kappa$ values of pure organic and inorganic particles were taken from the fitted lines to serve as a reference baseline, as shown in Figure S1. Based on the size-resolved $\kappa$-$f_{org}$ linear relationship (colored solid lines) and the measured $\kappa$ values (grey dashed line) of the SA-OOMs mixture, the corresponding $f_{org}$ values (colored dashed lines) for particles generated in the flow tube reactor were derived using Eq. 1"

[Figure]

Figure S1. (a) The measured $\kappa$ values of pure organic (OOMs) and inorganic (SA) under RH=20%. (b) Size-resolved $\kappa$-$f_{org}$ linear relationships (solid lines), taking 3, 5, 7, and 10 nm as examples. The grey dashed line represents the measured $\kappa$ of SA-OOMs particles. The colored dashed lines represent the derived $f_{org}$.

**5. Technical corrections:**

*Line 169: Differ from --> different from.*

*The given values in percent on lines 198 and 218 are too accurate. I would recommend using an accuracy of 1% or, in maximum, the accuracy of 0.1%.*

Corrected. These comments and suggestions have been rewritten in the revised manuscript.

**Refences:**

Cheng, Y., Su, H., Koop, T., Mikhailov, E., and Pöschl, U.: Size dependence of phase transitions in aerosol nanoparticles, Nat. Commun., 6, 5923, https://doi.org/10.1038/ncomms6923, 2015.

Poulain, L., Wu, Z., Petters, M. D., Wex, H., Hallbauer, E., Wehner, B., Massling, A., Kreidenweis, S. M., and Stratmann, F.: Towards closing the gap between hygroscopic growth and CCN activation for secondary organic aerosols – Part 3: Influence of the chemical composition on the hygroscopic properties and volatile fractions of aerosols, Atmos. Chem. Phys., 10, 3775–3785, https://doi.org/10.5194/acp-10-3775-2010, 2010.

Riipinen, I., Yli-Juuti, T., Pierce, J. R., Petäjä, T., Worsnop, D. R., Kulmala, M., and Donahue, N. M.: The contribution of organics to atmospheric nanoparticle growth, Nat. Geosci., 5, 453–458, https://doi.org/10.1038/ngeo1499, 2012.

Surdu, M., Lamkaddam, H., Wang, D. S., Bell, D. M., Xiao, M., Lee, C. P., Li, D., Caudillo, L., Marie, G., Scholz, W., Wang, M., Lopez, B., Piedehierro, A. A., Ataei, F., Baalbaki, R., Bertozzi, B., Bogert, P., Brasseur, Z., Dada, L., Duplissy, J., Finkenzeller, H., He, X.-C., Höhler, K., Korhonen, K., Krechmer, J. E., Lehtipalo, K., Mahfouz, N. G. A., Manninen, H. E., Marten, R., Massabò, D., Mauldin, R., Petäjä, T., Pfeifer, J., Philippov, M., Rörup, B., Simon, M., Shen, J., Umo, N. S., Vogel, F., Weber, S. K., Zauner-Wieczorek, M., Volkamer, R., Saathoff, H., Möhler, O., Kirkby, J., Worsnop, D. R., Kulmala, M., Stratmann, F., Hansel, A., Curtius, J., Welti, A., Riva, M., Donahue, N. M., Baltensperger, U., and El Haddad, I.: Molecular understanding of the enhancement in organic aerosol mass at high relative humidity, Environ. Sci. Technol., 57, 2297–2309, https://doi.org/10.1021/acs.est.2c04587, 2023.

---

## Author Comment (AC3)

*Response to Referee #2*

*General Comments*
*The manuscript by Zhang et al. discusses the organic contribution to nanoparticle growth based on laboratory measurements. Since direct composition measurements in this size range are difficult, there is still limited understanding of the relative roles of sulfuric acid and organics as a function of size. This study contributes to this understanding and is therefore suitable for publication in ACP after some revisions. The analysis and figures are presented very clearly, but the language should still be checked carefully.*

We thank the reviewer for the constructive suggestions and comments concerning our manuscript entitled "Unveiling the organic contribution to the initial particle growth in 3-10 nm size range" [egusphere-2025-4421]. We sincerely appreciate these insightful comments, which have significantly improved both the quality of our manuscript and the methodological rigor of our ongoing research.

We have prepared point-by-point responses to the reviewer's comments below. The original reviewer comments appear in italics. Our responses are presented in blue plain font, while the corresponding revisions in the manuscript appear in blue underlined font.

**Major comments**

*1. Comments and suggestions:*
*I am puzzled by the large systematic difference between the SA K-values in this study and earlier studies (Table 1). In the text and conclusions (r272), those are attributed to "the nano size effect and different measurement techniques". However, e.g. Kim et al. 2016 also looked at 10 nm particles, so the "nano size effect" cannot be the explanation there. The authors should thoroughly analyze and discuss the potential reasons for the differences.*

**Responses and Revisions:**
We thank the reviewer for this insightful comment. We agree that the large discrepancy, especially with the 10 nm data from Kim et al. (2016), warrants a more thorough discussion. In response to the reviewer's point, we acknowledge that the nano size effect alone cannot explain the difference at the specific 10 nm point. Our revised discussion now attributes the observed systematic differences to a combination of several factors:

(1) Fundamental difference in measurement techniques: We emphasize that the $\kappa$ values measured by our method (SFCPC based on supersaturated condition) and by the

method used in Kim et al. (2016) (nano-HTDMA based on subsaturated condition) are expected inherent different, where SA particles are expected to exhibit comparably higher hygroscopic growth at subsaturated conditions and lower CCN activity at supersaturation (Biskos et al., 2009; Massling et al., 2023). This is likely the primary source of the discrepancy.

(2) Persistent role of the nano size effect: While it may not explain the difference of 10 nm SA particles directly, the nano size effect is a critical factor that systematically lowers the $\kappa$ values across our entire measured size range (3-10 nm) compared to theoretical values for larger particles. Its influence is integral to understanding why our entire dataset is shifted to lower values.

(3) Synergistic effect and measurement scarcity: We think that the magnitude of the difference maybe due to a synergistic combination of the above factors. Furthermore, we highlight that the direct measurements in this size range are exceptionally rare. For SA particles, only Kim et al. (2016) reported directly measured $\kappa$ values as far as we know. The lack of a benchmark makes cross-technique comparisons challenging. This means that the discrepancies observed here hard to attribute to reasons other than instrument differences.

We have emphasized the method difference between this work and Kim et al. (2016) and provided further discussion on the nano-size effect and the potential contamination of $NH_3$ in Section 3.1:

"Hygroscopicity of SA particles also revealed a consistent decrease with rising RH: at RH = 80%, $\kappa$ values declined to 0.361 ± 0.013 for 3 nm and 0.171 ± 0.011 for 10 nm particles, decreasing 13% and 18% compared to RH = 20%. This demonstrates that SA particles exhibit reduced hygroscopicity under higher humidity conditions. Although bases such as ammonia/amine could in principle suppress the hygroscopicity of SA particles (Yishake et al., 2025), the $\kappa$ of an SA-ammonia/amine mixture would be expected to increase with RH, because a higher SA fraction (more acidic) is expected at higher RH (Chen et al., 2018). This $\kappa$-RH dependence is opposite to the trend observed in our study. Moreover, there was no identified source of bases in our system, so ammonia contamination can largely be ruled out. Furthermore, previous studies have shown that the average number of water molecules hydrating each $H_2SO_4$ molecule increases with RH (Kurtén et al., 2007; Temelso et al., 2012; Zollner et al., 2012). Therefore, the observed decrease in hygroscopicity can be attributed to the particles already containing more water at higher RH, resulting in a reduced capacity for additional water uptake.

The $\kappa$ values reported in previous studies are summarized in Table 1. The measured hygroscopicity results in this work are much lower than those reported in

previous studies (0.68-0.9, Petters and Kreidenweis, 2007; Shantz et al., 2008; Sullivan et al., 2010), which predicted by thermodynamic model in 30-80 nm based on the model parameters provided by Clegg et al. (1998). The hygroscopicity of newly formed sulfuric acid nanoparticles in CLOUD chamber was examined by a nano hygroscopicity tandem differential mobility analyser (nano-HTDMA) and the reported $\kappa$ values were $0.64 \pm 0.02$ and $0.52 \pm 0.02$ for 10 nm and 15 nm, respectively (Kim et al., 2016). The reported variation trend in particle size aligns with our findings but the $\kappa$ values were much larger than that in this work. While both Kim et al. (2016) and our study report $\kappa$ values of the oxidation products of $SO_2$, the measurement methods used are fundamentally different. Kim et al. (2016) employed a nano-HTDMA, which measures hygroscopic growth under subsaturated conditions. In contrast, our SFCPC method derives $\kappa$ values by activating particles under supersaturated conditions. As discussed in previous studies, the measurement condition can induce inherently different $\kappa$ values (Biskos et al., 2009; Massling et al., 2023). Specifically, SA particles are expected to exhibit comparably higher hygroscopic growth under subsaturated conditions and lower CCN activity at supersaturation. Therefore, this methodological difference is likely the primary factor contributing to this discrepancy in $\kappa$ values. In addition, for sub-10 nm particles, the enhanced Kelvin effect (compared to that for larger CCN-size particles) dramatically raises the energy barrier for vapor condensation, directly contributing to suppressed hygroscopicity. Furthermore, the potential size-dependent influence on phase state may reflect a tendency for nanoparticles to remain in a liquid or mixed phase (Cheng et al., 2015). These nano-size effects on the thermodynamic and physical properties of aerosol particles are likely significant contributing factors in explaining the distinctively lower $\kappa$ values observed in our study. As far as we know, direct measurements of the $\kappa$ of sulfuric acid particles remain limited in the literature, and our results is the first measurement under supersaturation condition in 3-10 nm range."

**2. Comments and suggestions:**
*Can there be ammonia and/or amine contamination in your system? If yes, how would that affect the results and would that be constant with RH?*

**Responses and Revisions:**

We thank the reviewer for raising this important point regarding potential ammonia/amine contamination and its influence. The possibility and implications have been carefully considered. Indeed, if present, ammonia (or amines) could affect our results by partially neutralizing the sulfuric acid ($H_2SO_4$, SA) particles, as acid-base reactions can alter aerosol hygroscopicity. Literature suggests that for mixtures like SA-

ammonia, the measured hygroscopic growth would be lower than the theoretical prediction for pure SA but higher than that of ammonium sulfate (Yishake et al., 2025). However, we argue that ammonia/amine contamination is unlikely to be the primary factor driving the observed decreasing hygroscopicity trend with increasing RH in our experiments, for the following reasons:

(1) Controlled experimental system: There was no deliberate or identified source of ammonia in the experimental setup. Most notably, the quartz flow tube was replaced for every experimental group to eliminate potential cross-contamination between runs. This practice significantly reduces the possibility of persistent ammonia residues affecting the results.

(2) Acidic nature of particles under trace contamination: Even if trace amounts of ammonia were present, the SA concentration in our system was overwhelmingly higher than of any potential ammonia contaminant. Previous study indicates that for sub-10 nm particles formed from SA with bases, the particles remain highly acidic when the ammonia is in substantial excess (Chen et al., 2018). Therefore, the potential minor contamination would likely result in particles whose composition and properties are still dominated by acidic SA rather than neutral salts.

(3) Inconsistency with expected RH dependence: This is the most critical point. If ammonia contamination were significant and driving the measured hygroscopicity suppression, one would expect the effective hygroscopicity parameter ($\kappa$) of the resulting partially neutralized particles to increase with increasing RH. This is because higher fraction of SA (more acidic) is expected at higher RH as discussed above, typically enhancing the hygroscopic response of SA-ammonia mixture. However, our observations show the opposite trend: a clear decrease in $\kappa$ with increasing RH, where this discrepancy suggests that the dominant mechanism causing the RH-dependent decrease in $\kappa$ in our data is unlikely to be an artifact of neutralization by base.

In conclusion, while we acknowledge the reviewer's valid hypothesis, the combination of our controlled experimental setup, the expected persistence of highly acidic particles, and—most decisively—the opposite RH trend compared to what a significant ammonia influence would predict, leads us to cautiously conclude that ammonia/amine contamination is not the main factor responsible for the observed decrease in particle hygroscopicity with increasing RH in this study.

We have added a brief sentence in the revised manuscript (Section 3.1):

"Although bases such as ammonia/amine could in principle suppress the hygroscopicity of SA particles (Yishake et al., 2025), the $\kappa$ of an SA-ammonia/amine mixture would be expected to increase with RH, because a higher SA fraction (more acidic) is expected at higher RH (Chen et al., 2018). This $\kappa$-RH dependence is opposite

to the trend observed in our study. Moreover, there was no identified source of bases in our system, so ammonia contamination can largely be ruled out."

*3. Comments and suggestions:*
*In row 138 you mention "the mass spectrometry" which was not discussed anywhere else. What mass spectrometer was used and for which part of the experiment?*

**Responses and Revisions:**

We thank the reviewer for pointing out the need for clarification. The mass spectrometry analysis was not a core component of the main experiment but was employed as a preliminary, auxiliary measurement to rule out potential interference from organic compounds in the observed hygroscopic behavior of sulfuric acid. Specifically, an iodide chemical ionization mass spectrometer (Vocus AIM) was used to characterize the composition of gaseous oxidation products prior to conducting the core experiments. The corresponding results are now provided as Figure S2 in the Supplementary Information. The results confirmed the absence of significant organic signals, supporting the conclusion that organic contamination did not drive the decreasing trend mentioned in the text. We have accordingly revised the manuscript in Section 3.1 to ensure clarity:

"Moreover, prior to the hygroscopicity measurements, the oxidation products of $SO_2$ were characterized using an iodide chemical ionization mass spectrometry (Vocus AIM), which detected no significant organic compound signals in the gas phase (Figure S2). Therefore, this decreasing trend cannot be attributed to organic contamination."

[Figure]

Figure S2. Representative Vocus AIM negative ion mass spectra of the oxidation products of $SO_2$.

**Minor comments**

*4. Comments and suggestions:*
*What is the RH of the DMA sheat flow?*

**Responses and Revisions:**

  We sincerely thank the reviewer for raising this important point. In the present study, the RH of the DMA sheath flow was not measured. As the reviewer rightly implies, this parameter is critically important in HTDMA-based hygroscopicity measurements, where precise control of the sheath flow RH is essential for accurate determination of the particle growth factor. However, for measurement techniques analogous to ours, which utilize CCNC and determine hygroscopicity by activating particles under supersaturation conditions, the RH of DMA sheath flow is generally not monitored or reported (Roberts and Nenes, 2005; Rose et al., 2008; Wang et al., 2015). Therefore, in line with this established methodology, our experimental design did not specifically focus on the RH of the DMA sheath flow. Nonetheless, we fully agree with the reviewer that direct measurement of this parameter could provide valuable additional information and further improve the reliability of particle sizing, particularly in ensuring the consistency of the dry diameter selection. In future work, we will

incorporate in situ RH monitoring of the sheath flow to enable more precise characterization and enhance methodological rigor. Thank you again for this valuable and constructive suggestion.

*5. Comments and suggestions:*
*Abstract, row 21: more pronounced compared to what?*

**Responses and Revisions:**

We sincerely apologize for this unclear expression. We have revised it more clearly:

"Compared to 3-5 nm, this enhancement was more pronounced for 5-10 nm particles associated with the incorporation of increased yields of more volatile oxidation products and Kelvin effect."

**Refences:**

Biskos, G., Buseck, P. R., and Martin, S. T.: Hygroscopic growth of nucleation-mode acidic sulfate particles, J. Aerosol Sci., 40, 338–347, https://doi.org/10.1016/j.jaerosci.2008.12.003, 2009.

Chen, H., Chee, S., Lawler, M. J., Barsanti, K. C., Wong, B. M., and Smith, J. N.: Size resolved chemical composition of nanoparticles from reactions of sulfuric acid with ammonia and dimethylamine, Aerosol Sci. Technol., 52, 1120–1133, https://doi.org/10.1080/02786826.2018.1490005, 2018.

Cheng, Y., Su, H., Koop, T., Mikhailov, E., and Pöschl, U.: Size dependence of phase transitions in aerosol nanoparticles, Nat. Commun., 6, 5923, https://doi.org/10.1038/ncomms6923, 2015.

Clegg, S. L., Brimblecombe, P., and Wexler, A. S.: Thermodynamic Model of the System $H^+$ $-NH_4^+$ $-Na^+$ $-SO_4^{2-}$ $-NO_3^-$ $-Cl^-$ $-H_2O$ at 298.15 K, J. Phys. Chem. A, 102, 2155–2171, https://doi.org/10.1021/jp973043j, 1998.

Kim, J., Ahlm, L., Yli-Juuti, T., Lawler, M., Keskinen, H., Tröstl, J., Schobesberger, S., Duplissy, J., Amorim, A., Bianchi, F., Donahue, N. M., Flagan, R. C., Hakala, J., Heinritzi, M., Jokinen, T., Kürten, A., Laaksonen, A., Lehtipalo, K., Miettinen, P., Petäjä, T., Rissanen, M. P., Rondo, L., Sengupta, K., Simon, M., Tomé, A., Williamson, C., Wimmer, D., Winkler, P. M., Ehrhart, S., Ye, P., Kirkby, J., Curtius, J., Baltensperger, U., Kulmala, M., Lehtinen, K. E. J., Smith, J. N., Riipinen, I., and Virtanen, A.: Hygroscopicity of nanoparticles produced from homogeneous nucleation in the CLOUD experiments, Atmos. Chem. Phys., 16, 293–304, https://doi.org/10.5194/acp-16-293-2016, 2016.

Kurtén, T., Noppel, M., Vehkamäki, H., Salonen, M., and Kulmala, M.: Quantum chemical studies of hydrate formation of H2SO4 and HSO4– - ProQuest, Boreal Environ. Res., 2007.

Massling, A., Lange, R., Pernov, J. B., Gosewinkel, U., Sørensen, L.-L., and Skov, H.: Measurement report: High arctic aerosol hygroscopicity at sub- and supersaturated conditions during spring and summer, Atmos. Chem. Phys., 23, 4931–4953, https://doi.org/10.5194/acp-23-4931-2023, 2023.

Petters, M. D. and Kreidenweis, S. M.: A single parameter representation of hygroscopic growth and cloud condensation nucleus activity, Atmos. Chem. Phys., 7, 1961–1971, https://doi.org/10.5194/acp-7-1961-2007, 2007.

Roberts, G. C. and Nenes, A.: A continuous-flow streamwise thermal-gradient CCN chamber for atmospheric measurements, Aerosol Science and Technology, 39, 206–221, https://doi.org/10.1080/027868290913988, 2005.

Rose, D., Gunthe, S. S., Mikhailov, E., Frank, G. P., Dusek, U., Andreae, M. O., and Pöschl, U.: Calibration and measurement uncertainties of a continuous-flow cloud condensation nuclei counter (DMT-CCNC): CCN activation of ammonium sulfate and sodium chloride aerosol particles in theory and experiment, Atmos. Chem. Phys., 8, 1153–1179, https://doi.org/10.5194/acp-8-1153-2008, 2008.

Shantz, N. C., Leaitch, W. R., Phinney, L., Mozurkewich, M., and Toom-Sauntry, D.: The effect of organic compounds on the growth rate of cloud droplets in marine and forest settings, Atmos. Chem. Phys., 8, 5869–5887, https://doi.org/10.5194/acp-8-5869-2008, 2008.

Sullivan, R. C., Petters, M. D., DeMott, P. J., Kreidenweis, S. M., Wex, H., Niedermeier, D., Hartmann, S., Clauss, T., Stratmann, F., Reitz, P., Schneider, J., and Sierau, B.: Irreversible loss of ice nucleation active sites in mineral dust particles caused by sulphuric acid condensation, Atmos. Chem. Phys., 10, 11471–11487, https://doi.org/10.5194/acp-10-11471-2010, 2010.

Temelso, B., Morrell, T. E., Shields, R. M., Allodi, M. A., Wood, E. K., Kirschner, K. N., Castonguay, T. C., Archer, K. A., and Shields, G. C.: Quantum mechanical study of sulfuric acid hydration: Atmospheric implications, J. Phys. Chem. A, 116, 2209–2224, https://doi.org/10.1021/jp2119026, 2012.

Wang, Z., Su, H., Wang, X., Ma, N., Wiedensohler, A., Pöschl, U., and Cheng, Y.: Scanning supersaturation condensation particle counter applied as a nano-CCN counter for size-resolved analysis of the hygroscopicity and chemical composition of nanoparticles, Atmos. Meas. Tech., 8, 2161–2172, https://doi.org/10.5194/amt-8-2161-2015, 2015.

Yishake, J., Zang, H., Tan, R., Yao, L., Guo, S., Zhao, Y., and Li, C.: Size- and Composition-Dependent Hygroscopic Growth of Sub-20 nm Atmospherically Relevant Particles: Implications for New Particle Survival, Environ. Sci. Technol., acs.est.5c00068, https://doi.org/10.1021/acs.est.5c00068, 2025.

Zollner, J. H., Glasoe, W. A., Panta, B., Carlson, K. K., McMurry, P. H., and Hanson, D. R.: Sulfuric acid nucleation: Power dependencies, variation with relative humidity, and effect of bases, Atmos. Chem. Phys., 12, 4399–4411, https://doi.org/10.5194/acp-12-4399-2012, 2012.

---

## Author Comment (AC4)

**Response to Referee #3**

**General Comments**
*Zhang et al measured the hygroscopicity of aerosol particles containing sulfuric acid (SA), α-pinene-derived oxygenated organic molecules (OOMs), and their mixture in the size range of 3-10 nm. The measurement was conducted using a custom-designed scanning flow condensation particle counter (SFCPC). By doing this, the organic mass fraction for the mixture particle can be derived for the SA-OOM mixture particles that were generated at different SA/α-pinene ratios and humidity. The main claim is that OOMs' contribution to the particle phase becomes increasingly important as particle size increases. When the motivation of the study and the use of language are overall good, the current level of analysis is not convincing enough to back up the main claim. If the following comments could be adequately addressed, I would suggest considering the work for publication.*

We thank the reviewer for the constructive suggestions and comments concerning our manuscript entitled "Unveiling the organic contribution to the initial particle growth in 3-10 nm size range" [egusphere-2025-4421]. We sincerely appreciate these insightful comments, which have significantly improved both the quality of our manuscript and the methodological rigor of our ongoing research.

We have prepared point-by-point responses to the reviewer's comments below. The original reviewer comments appear in italics. Our responses are presented in blue plain font, while the corresponding revisions in the manuscript appear in blue underlined font.

**Major comments**

**1. Comments and suggestions:**
*Novelty of the work: The SFCPC has been used to study the CCN activity of nanoparticles several times before. The authors must provide a summary of their previous work (Wang et al., 2015; Zhang et al., 2023; Zhang et al., 2025) based on the SFCPC in the introduction. Unfortunately, the authors barely mention their previous work using SFCPC, and therefore, readers most likely have no clue about the novelty of the current work.*

**Responses and Revisions:**
We thank the reviewer for the constructive suggestions and comments. This study marks the first application of the SFCPC to infer the chemical composition of mixtures containing unknown components. Our previous works (Wang et al., 2015; Zhang et al., 2023, 2025) focused on the development and characterization of novel instrumental

methods, along with the assessment of associated uncertainties, and did not encompassed investigations into the generation and growth of new particles. The novelty of this work and its distinct differences from our previous studies are summarized as follows:

(1) Wang et al. (2015) and Zhang et al. (2023) developed and characterized the nano-CCNC and SFCPC, respectively, where the major difference between these two studies is the method used to alter the inner supersaturation. The linear relationship of $\kappa$-$f_{org}$ were established between ammonium sulphate (AS) and levoglucosan/sucrose generated by Electrospray from solutions with known mixing ratios, to demonstrate its ability to infer chemical composition.

(2) Zhang et al. (2025) focused on the modelling studies of exemplary substances to assess uncertainties arising from the dependence of $\kappa$ on particle size and thermodynamic models. Although it discussed chemical composition inference (Section 3.4), its primary focus was on verifying the linear relationship of $\kappa$-$f_{org}$ and assessing the associated uncertainties. More critically, that work also utilized particles generated via Electrospray from solutions of AS and *cis*-pinonic acid. This experimental approach, particularly the use of Electrospray to generate particles from solution, could not directly represent the gas-phase precursor mechanisms that govern atmospheric nucleation and growth processes.

(3) The present study, by contrast, is designed to directly probe NPF and subsequent growth by simulating atmospheric processes. Utilizing a flow tube reactor, we introduced gas-phase precursors ($SO_2$ and $\alpha$-pinene) to study nucleation and particle growth under controlled conditions. This methodology enables us to demonstrate the role of their oxidation products in the growth process. Thus, our work provides valuable and novel physical and chemical results of nanoparticle down to 3 nm that have not been previously reported, thereby yielding critical implications for understanding atmospheric particle formation and growth.

We agree with the reviewer that highlighting the distinction between this work and our previous studies is crucial for underscoring its novelty. We acknowledge that our initial manuscript did not sufficiently articulate this progression. The differences in research objectives and contents (applied compositional analysis vs. instrumental development and characterization) led us to primarily cite prior work in the Introduction and Methods sections as foundational references, rather than providing a focused comparative discussion in the Results section. Accordingly, in the revised manuscript, we have added relevant text to summary our previous work (Section 2.3):

"Our previous work extended the linear relationship between $\kappa$ and $f_{org}$ to 3-10 nm size range for AS and levoglucosan/sucrose mixed particles, which were generated via

Electrospray from solutions with known mixing ratios (Wang et al., 2015; Zhang et al., 2023). Zhang et al. (2025) assessed the uncertainty arising from the dependence of $\kappa$ on particle size, also based on Electrospray-generated particles (AS and *cis*-pinonic acid). In this work, we introduced $\kappa$-$f_{org}$ linear relationship into the mixing products from flow tube—a setup designed to simulate atmospheric processes—in order to explore the organic content in the particulate phase. Furthermore, we established size-resolved $\kappa$-$f_{org}$ linear relationship to eliminate the mentioned uncertainty. The application of the $\kappa$-$f_{org}$ linear relationship relies on the assumption of ideal internal mixing within the particles. Under our experimental conditions, for in situ freshly formed 3-10 nm particles, the characteristic mixing times are short, and organic-inorganic mixtures are likely to remain liquid and well-mixed (Cheng et al., 2015). Therefore, the ideal internal mixing assumption is reasonable. For SA-OOMs mixture, we use the $\kappa$ values of pure organic (OOMs) and inorganic (SA) experiment groups to represent the hygroscopicity of the organic and inorganic component in the mixture particles, respectively. Although the organic and inorganic components in mixture may not be identical to those in pure organic and inorganic particles of the same size under the same RH due to the potential change of oxidation processes, this simplification is a necessary given the current inability to directly measure the composition and component-specific hygroscopicity of particles in the 3-10 nm size range. To further reduce uncertainties associated with the linear relationship, the $\kappa$ values of pure organic and inorganic particles were taken from the fitted lines to serve as a reference baseline, as shown in Figure S1. Based on the size-resolved $\kappa$-$f_{org}$ linear relationship (colored solid lines) and the measured $\kappa$ values (grey dashed line) of the SA-OOMs mixture, the corresponding $f_{org}$ values (colored dashed lines) for particles generated in the flow tube reactor were derived using Eq. 1"

**2. Comments and suggestions:**
*Aerosol Generation: There are a couple of questions that need clarification with support from compositional measurement and/or modelling.*

We sincerely appreciate the reviewer's constructive comments regarding the aerosol generation. Our responses to each sub-point are provided below.

**2.1 What was the estimated OH concentration in the flow tube per experiment?**

**Responses and Revisions:**

Thank you for your comment. Hydroxyl radicals (OH·) would participate in the oxidation processes of both $SO_2$ and α-pinene, but its concentration was not estimated in the present experimental setup. In this study, we focus primarily on the apparent

hygroscopicity ($\kappa$) and organic mass fraction ($f_{org}$) of the formed particles, rather than on molecular-level details or specific chemical pathways. Moreover, in the mixed $SO_2$/α-pinene system used here, determining the OH· concentration for each experiment is particularly complex. Although our current experimental setup and instrumentation do not allow for reliable estimation of OH· concentrations, this limitation does not affect the measurement of apparent hygroscopicity or the subsequent determination of organic content for the newly formed particles, which are the central objectives of the present work.

We have added the following explicit statement in Section 2.1:

"While the present study did not estimate the OH· concentration and further elucidate specific mechanistic pathways, this simplification is justified because, to the best of our knowledge, no existing studies have clearly demonstrated significant differences in the hygroscopic performance of pure OOMs derived from different α-pinene oxidation pathways."

**2.2 How was α-pinene consumed via ozonolysis and OH oxidation pathways?**

**Responses and Revisions:**

We thank the Referee for this insightful comment. As noted in the original manuscript, α-pinene was simultaneously exposed to $O_3$ and OH· radicals due to the inevitable production of OH· from $O_3$ photolysis and subsequent reactions.

The two oxidation pathways are generally understood as follows: (1) The gas-phase ozonolysis of α-pinene proceeds via the formation of Criegee intermediates (carbonyl oxides; CI), which undergo rapid unimolecular reactions to form vinyl hydroperoxides (VHP) and subsequently different peroxy radical ($RO_2$) isomers, leading to low-volatility oxygenated organic molecules (OOMs) (Iyer et al., 2021; Yang et al., 2025). (2) $O_3$ undergoes photolysis to produce $O(^1D)$ atoms, which then react with water vapor to generate OH· radicals. OH-initiated oxidation of α-pinene proceeds mainly via OH· addition to the double bond, forming peroxy radicals ($RO_2$) (Berndt et al., 2016; Kang et al., 2025). These $RO_2$ radicals undergo autoxidation and $HO_2$ cycling, producing OOMs that could contribute to the growth of particles.

The present study, however, primarily focuses on the bulk physicochemical properties (hygroscopicity and organic fraction) of the 3-10 nm particles generated under known oxidation conditions, rather than on elucidating specific mechanistic pathways. Furthermore, to the best of our knowledge, no existing studies have clearly indicated significant differences in the hygroscopicity of pure OOMs derived from different α-pinene oxidation pathways. The methods employed in this study do not

provide molecular composition information, which would be required for pathway assignment. Should future techniques enable molecular speciation of such small particles, this would greatly advance our understanding of nucleation and growth mechanisms. We appreciate the reviewer's comment, which has helped us to clarify the scope of our work.

In response to this comment, we have added the following explicit statement in Section 2.1:

"For pure organic experimental groups, OOMs were produced from organic peroxides formed from oxidation reactions of α-pinene (Kirkby et al., 2016; Lee et al., 2019). α-pinene was exposed to ozone and also to hydroxyl radicals (OH·) due to the unavoidable production of OH· from ozone photolysis and secondary reactions. Although detailed molecular composition information could not obtained in our work, the ozonolysis pathway is generally understood to proceed via Criegee intermediates, leading to various peroxy radicals and subsequent low-volatility products (Iyer et al., 2021; Yang et al., 2025). Similarly, OH-initiated oxidation proceeds mainly via OH· addition, forming peroxy radicals that further react to produce condensable organic species (Berndt et al., 2016; Kang et al., 2025). While the present study did not estimate the OH· concentration and further elucidate specific mechanistic pathways, this simplification is justified because, to the best of our knowledge, no existing studies have clearly demonstrated significant differences in the hygroscopic performance of pure OOMs derived from different α-pinene oxidation pathways."

*2.3 Was the composition of OOM comparable between different sizes for the OOM-only particles?*

**Responses and Revisions:**

We thank the reviewer for raising this insightful and important question regarding the potential size-dependent composition of the nanoparticles formed from oxidation products of α-pinene.

Based on the governing physical principles of gas-to-particle partitioning, the chemical composition of OOMs particles formed from α-pinene oxidation products is not expected to be strictly identical across different sizes. The partitioning of organic vapors into the particle phase is influenced by their volatility and the Kelvin effect (Stolzenburg et al., 2023), which becomes increasingly significant for smaller particles, potentially altering the species participating in nanoparticle growth. Therefore, some size-dependent variation in the detailed chemical composition is likely. Despite this expected variation, current analytical techniques also pose limitations for obtaining the

exact composition of newly formed 3-10 nm particles (Smith et al., 2021; Zhang et al., 2022).

Meanwhile, our key experimental observation is that the $\kappa$ of these pure OOMs particles showed no significant dependence on particle size within the 3-10 nm range (as shown in Figure 2b). This suggests that the effective average hygroscopicity of the mixture of OOMs condensing onto particles in this size range remains relatively constant. This finding is consistent with the observation by Frosch et al. (2011) of an almost constant $\kappa$ for 50-150 nm particles from α-pinene oxidation. It seems to indicate that either the composition shifts involve compounds with similar $\kappa$ values, or that the net effect of compositional changes on the bulk hygroscopicity is minor within this specific diameter range. To our knowledge, no studies have specifically examined the particle size dependence of hygroscopicity for 3-10 nm particles from α-pinene oxidation. In our analysis, we have incorporated the potential size-dependent variation in chemical composition by establishing independent linear relationships for each particle diameter, thereby eliminating potential uncertainties in the inferred organic mass fraction arising from size-dependent differences.

We have therefore revised the manuscript to include this point explicitly in the Section 3.1:

"Different from the SA results, the OOMs particles formed by the oxidation products of α-pinene have almost constant $\kappa$ values in the 3-10 nm diameter range. As shown in Figure 2(b), $\kappa$ of the pure organic group under 20% RH is ~0.065. This result indicates that the OOMs particles exhibit non-hygroscopicity compared to SA particles and its hygroscopicity has no significant dependence on particle diameter. While the detailed chemical composition of OOMs in these nanoparticles may vary with size due to volatility-dependent partitioning and the Kelvin effect, the observed constancy in $\kappa$ suggests that the effective hygroscopic properties of the condensing mixture do not change significantly within this size range. This finding is consistent with Frosch et al. (2011), who also observed almost constant $\kappa$ values (0.11 ± 0.02) for 50-150 nm particles from α-pinene oxidation. However, to our knowledge, research on particle size dependence is very rare, and no studies have specifically examined the hygroscopicity of α-pinene oxidation particles in the 3-10 nm size range. Alfarra et al. (2013) measured the hygroscopicity of particles formed from α-pinene oxidation via both OH radicals and $O_3$ using CCNC. Our results align closely with their reported $\kappa$ values (0.1-0.16), which were also observed under supersaturation conditions. Furthermore, numerous studies have investigated the hygroscopicity of α-pinene oxidation products using CCNC or HTDMA, reporting a wide range of $\kappa$ values (0.03-0.19, Cain et al., 2021; Duplissy et al., 2011; Engelhart et al., 2008; Massoli et al., 2010; Razafindrambinina et

al., 2022; Wang et al., 2019; Zhao et al., 2016). All experimental results reported here fall within the established range of values found in existing publications."

**2.4** *How can the authors be sure the composition of OOM was the same between the OOM-only particles and SA-OOM particles for a specific size under the same RH?*

**Responses and Revisions:**

We thank the reviewer for raising this important point regarding the compositional consistency of the OOMs in different experimental groups. We fully agree that this is an important assumption in our analysis, and we appreciate the opportunity to clarify our rationale.

In this study, we aimed to derive insights into the organic contribution to the initial particle growth in 3-10 nm size range by utilizing the reported linear relationship between hygroscopicity ($\kappa$) and the organic mass fraction ($f_{org}$), where direct measurement of particulate-phase composition is extremely challenging. When applying this $\kappa$-$f_{org}$ linear relationship to infer the organic fraction in SA-OOMs particles, a representative hygroscopicity parameter for the organic component ($\kappa_{org}$) is required. In the absence of direct measurements of $\kappa$ for the organics in the mixed SA-OOMs particles, we used the $\kappa$ value measured for pure OOMs particles as the best available value for $\kappa_{org}$ in the mixtures.

We acknowledge that the organic composition in mixed SA-OOMs particles may not be identical to that in pure OOMs particles for the same particle size under the same RH. The presence of $SO_2$ and sulfuric acid (SA) could potentially alter the oxidation and condensation pathways. However, it is currently unclear how it affects the composition of organic composition and the hygroscopic performance of organic parts in such small particles. Moreover, the direct measurement of the separate $\kappa$ values of organics in mixture particles is not currently feasible. Therefore, using the measured $\kappa$ of pure OOMs particles represents the most reasonable and practical approximation available to apply the $\kappa$-$f_{org}$ linear relationship to our mixed system. This approach allows us to provide valuable, albeit indirectly-informed, insights into the early growth of atmospheric nanoparticles.

We agree that future measurements capable of characterizing the detailed molecular composition and component-specific hygroscopicity of nanoparticles in this size range would be invaluable. Such data would significantly advance the understanding of the nanoparticle growth and refine the application of hygroscopicity-based approach. We have added related text in the revised manuscript to highlight this point and the associated assumption more explicitly:

Section 2.3: "For SA-OOMs mixture, we use the $\kappa$ values of pure organic (OOMs) and inorganic (SA) experiment groups to represent the hygroscopicity of the organic and inorganic component in the mixture particles, respectively. Although the organic and inorganic components in mixture may not be identical to those in pure organic and inorganic particles of the same size under the same RH due to the potential change of oxidation processes, this simplification is a necessary given the current inability to directly measure the composition and component-specific hygroscopicity of particles in the 3-10 nm size range."

**3. Comments and suggestions:**

*Determination of forg: It looks like the eq (1) is based on the assumption of ideal mixing of SA and OOM. If so, can you discuss the uncertainty regarding this assumption and how it will impact the determination of forg?*

**Responses and Revisions:**

We thank the reviewer for raising this critical point regarding the assumption underlying Eq. (1). We fully agree that the $\kappa$-$f_{\text{org}}$ linear relationship is based on the assumption of ideal internal mixing between sulfuric acid (SA) and oxidized organic molecules (OOMs). Below, we discuss the potential uncertainties associated with this assumption and their impact on the $f_{\text{org}}$ determination.

As the reviewer rightly noted, deviations from ideal internal mixing—such as external mixing, core-shell morphology, or liquid-liquid phase separation (LLPS)— could cause the linear mixing rule to break down. In an extreme core-shell scenario where a pure SA core is fully coated by OOMs, the overall particle hygroscopicity would be governed primarily by the organic shell. Consequently, the measured $\kappa_{\text{mix}}$ of SA-OOMs would approach the value of $\kappa_{\text{org}}$, leading to a substantial overestimation of $f_{\text{org}}$ (approaching ~100%, in contrast to the actual organic fraction of SA-OOMs). While we recognize these potential biases, quantitatively describing the $\kappa$-$f_{\text{org}}$ relationship under every possible non-ideal mixing state remains challenging, due to the absence of a generalized theoretical framework and direct observational constraints for particles in this size range.

Nevertheless, we argue that the ideal internal mixing assumption is reasonable for our experimental conditions (in situ freshly formed 3-10 nm particles). For nanoparticles of this scale, characteristic mixing times are short, and organic-inorganic mixtures are likely to remain liquid and well-mixed (Cheng et al., 2015), thereby minimizing the prevalence of phase-separated or core-shell structures. Thus, although we cannot fully exclude morphological uncertainties, the $\kappa$-$f_{\text{org}}$ linear relationship provides a well-established approach to estimate composition from hygroscopicity in

such systems (Dusek et al., 2010; Pöhlker et al., 2023; Vogel et al., 2016; Zhou et al., 2024).

To ensure clarity, we have revised the manuscript (Section 2.3) to explicitly state this key assumption:

"In this work, we introduced $\kappa$-$f_{\mathrm{org}}$ linear relationship into the mixing products from flow tube—a setup designed to simulate atmospheric processes—in order to explore the organic content in the particulate phase. Furthermore, we established size-resolved $\kappa$-$f_{\mathrm{org}}$ linear relationship to eliminate the mentioned uncertainty. The application of the $\kappa$-$f_{\mathrm{org}}$ linear relationship relies on the assumption of ideal internal mixing within the particles. Under our experimental conditions, for in situ freshly formed 3-10 nm particles, the characteristic mixing times are short, and organic-inorganic mixtures are likely to remain liquid and well-mixed (Cheng et al., 2015). Therefore, the ideal internal mixing assumption is reasonable."

**Minor comments**

**4. Comments and suggestions:**
*Lines 114 – 116: The description for the use of the electrometer needs to be clarified. Was it used to count total particle concentrations?*

**Responses and Revisions:**

We thank the reviewer for this helpful comment. Yes, the electrometer was used to measure the total particle concentration ($N_{\mathrm{total}}$) in our experimental setup. To clarify this in the text, we have revised the description in Section 2.2 accordingly. The revised text now explicitly states that the electrometer was operated in parallel to obtain $N_{\mathrm{total}}$, which serves as the reference for calculating the counting efficiency:

"SFCPC, which was improved from a water-based condensation particle counter (WCPC, TSI model 3788) and could change the supersaturation fast by altering the sample flow, was deployed to count the activated particle concentration ($N$). And an electrometer (EM, TSI model 3068B) was operated in parallel to measure the total particle concentration ($N_{\mathrm{total}}$), where the counting efficiency of sampled particles ($f_{\mathrm{count}}$) in different supersaturation ($S$) conditions could be obtained by $N/N_{\mathrm{total}}$."

**5. Comments and suggestions:**
*Line 138: Where is the mass spectrometry data?*

**Responses and Revisions:**

Thank you for your comment. The mass spectrometry data from the iodide chemical ionization mass spectrometry (Vocus AIM) are now provided as Figure S2 in the Supplementary Information, and we have added the corresponding citation in the

revised manuscript. These data were obtained prior to the main experiments as an auxiliary characterization to confirm the absence of significant organic compound signals in the gas phase, thereby excluding organic contamination as a driver for the observed trend. The revised text in Section 3.1 is as follows:

"Moreover, prior to the hygroscopicity measurements, the oxidation products of $SO_2$ were characterized using an iodide chemical ionization mass spectrometry (Vocus AIM), which detected no significant organic compound signals in the gas phase (Figure S2). Therefore, this decreasing trend cannot be attributed to organic contamination."

[Figure]

Figure S2. Representative Vocus AIM negative ion mass spectra of the oxidation products of $SO_2$.

**6. Comments and suggestions:**
*Line 144: If SA exist in the form as (H2SO4)m-(H2O)n, how will the m and n change regarding particle size and the presence of OOM?*

**Responses and Revisions:**

We thank the reviewer for raising this important point regarding the evolution of the composition, expressed as the stoichiometric coefficients m and n in $(H_2SO_4)_m$-$(H_2O)_n$, with particle size and in the presence of OOMs. Our response addresses each aspect below:

(1) Dependence of m/n on particle size: As discussed in the manuscript, the sulfuric acid and water binary nucleation is the basic mechanism for the SA particle

formation, the water molecules play a significant role through hydration (Kulmala et al., 1998; Lee et al., 2019; Stolzenburg et al., 2023; Yu et al., 2017). The initial formation of $H_2SO_4 \cdot H_2O$ molecular clusters is followed by rapid addition of further $H_2O$ molecules and these stepwise hydrates process ultimately leads to particles covered with water during growth (Couling et al., 2003; Matsubara et al., 2009), this mean the m/n would decrease with the increasing size. However, we are unable to determine specific m or n values through experiments, and we have not found data of them for 3-10 nm sulfuric acid particles in the literature.

(2) Influence of OOMs on m and n: OOMs, formed from α-pinene oxidation, may influence SA particle formation through several ways, such as participating in cluster/particle formation themselves or altering the oxidative environment for $SO_2$ (e.g., via Criegee intermediates or OH radical). While these interactions would potentially affect the overall composition and growth dynamics of the mixed SA-OOMs particles, molecular-level quantification of the particle composition or determination of m and n values in $(H_2SO_4)_m$-$(H_2O)_n$ remains a significant experimental challenge for both our approach and other analytical methods.

(3) Rationale for our analytical approach: Given the above constraint, a core assumption in our analysis is necessary: we use the hygroscopicity derived from the pure inorganic system (SA from $SO_2$ oxidation) as a proxy for the hygroscopicity of the inorganic fraction within the mixed SA-OOMs particles. This implies that, we assume the effective m and n of the $(H_2SO_4)_m$-$(H_2O)_n$ component are similar in both pure and mixed systems with same particle size. This assumption allows us to quantify the organic mass fraction ($f_{org}$) of overall particle and the contribution of organics to the particle initial growth, which is a primary objective of this work.

We fully agree with the reviewer that determining how m and n values of $(H_2SO_4)_m$-$(H_2O)_n$ evolve under various atmospheric conditions, including in complex mixtures with organics, is crucial for a mechanistic understanding of new particle growth. Advanced mass spectrometric techniques and coupled theoretical modeling focused on multicomponent clusters are needed to address this challenge. Insights from such future studies would greatly refine growth models and our ability to predict aerosol climate impacts.

We have added related text in the revised manuscript to highlight this simplification more explicitly:

Section 2.3: "For SA-OOMs mixture, we use the $\kappa$ values of pure organic (OOMs) and inorganic (SA) experiment groups to represent the hygroscopicity of the organic and inorganic component in the mixture particles, respectively. Although the organic and inorganic components in mixture may not be identical to those in pure organic and

inorganic particles of the same size under the same RH due to the potential change of oxidation processes, this simplification is a necessary given the current inability to directly measure the composition and component-specific hygroscopicity of particles in the 3-10 nm size range."

***Technical comments***

***7. Comments and suggestions:***
*Was a double-charge correction applied when deriving S from the measurement data?*

**Responses and Revisions:**

Thank you for raising this point. As noted in previous studies (Fuchs, 1963; Wiedensohler et al., 1986; Wiedensohler and Fissan, 1988), the probability of double-charge is extremely low for sub-20 nm particles and can be considered negligible. In our experiments, the calibration aerosols did not exceed 20 nm, and the size range of interest (3-10 nm) lies well within this limit. Therefore, we think that the influence of double-charge on the derived results is negligible. For clarity, a brief statement has been added in the revised manuscript (Section 2.2) indicating that no double-charge correction was applied for the studied size range.

"The aerosol particles were passed through a neutralizer (X-ray, TSI model 3080), and a nano-differential mobility analyzer (nano-DMA, TSI model 3086) was used to select charged monodisperse particles in diameter range of 3-10 nm (with intervals of 0.2 nm in the 3-4 nm range and intervals of 0.5 nm in the 4-10 nm range). Considering the negligible probability of double-charge for particles below 20 nm (Fuchs, 1963; Wiedensohler et al., 1986; Wiedensohler and Fissan, 1988), no double-charge correction was applied in the studied size range."

***8. Comments and suggestions:***
*What was the value for surface tension used in the equation to derive kappa?*

**Responses and Revisions:**

We thank the reviewer for this question. The $\kappa$-Köhler equation was applied following the specific implementation described by Equation 2 in Zhang et al. (2023). In the application, the value used for surface tension is that of pure water, $0.072 \text{ N m}^{-1}$. This has been explicitly clarified in the revised manuscript (Section 2.2):

"Then the $\kappa$ values could be obtained from dry diameter ($D_d$) and its corresponding $S$ based on $\kappa$-Köhler equation (Petters and Kreidenweis, 2007). The equation was

applied following Equation 2 in Zhang et al. (2023), in which the surface tension of water (0.072 N m$^{-1}$) was used."

**9. Comments and suggestions:**
*Line 161: Please clarify what the nano-size effect is.*

**Responses and Revisions:**

We thank the reviewer for raising this important question. In the context of our study, the "nano-size effect" specifically refers to the profound changes in particle behavior within the 3-10 nm size range, which is the focus of our work. These changes, driven by the reduction in particle size, primarily manifest as an enhanced Kelvin effect and potential alterations in phase state, both of which could influence hygroscopicity.

(1) The enhanced Kelvin effect: This is a fundamental thermodynamic principle wherein the equilibrium vapor pressure over a curved surface increases exponentially with its curvature. For nanoparticles (<10 nm), the extreme curvature dramatically elevates the energy barrier for vapor condensation. This directly suppresses water uptake, leading to a lower measured hygroscopicity parameter ($\kappa$).

(2) The size-dependent influence on phase state: particle size can fundamentally impact phase state, exhibiting a tendency of nanoparticles for staying in liquid and mixed phase. This potential liquid state can significantly alter physicochemical properties relevant to hygroscopicity, such as surface tension, viscosity, and solute dissolution kinetics, thereby further modifying the water uptake behavior.

Therefore, in our study, the term "nano-size effect" encapsulates the combined action of these two factors. Together, they provide a coherent explanation for the distinctively lower $\kappa$ we observed.

We have revised the relevant discussion in section 3.1 to incorporate this clarified of the nano-size effect:

"In addition, for sub-10 nm particles, the enhanced Kelvin effect (compared to that for larger CCN-size particles) dramatically raises the energy barrier for vapor condensation, directly contributing to suppressed hygroscopicity. Furthermore, the potential size-dependent influence on phase state may reflect a tendency for nanoparticles to remain in a liquid or mixed phase (Cheng et al., 2015). These nano-size effects on the thermodynamic and physical properties of aerosol particles are likely significant contributing factors in explaining the distinctively lower $\kappa$ values observed in our study."

**Refences:**

Alfarra, M. R., Good, N., Wyche, K. P., Hamilton, J. F., Monks, P. S., Lewis, A. C., and McFiggans, G.: Water uptake is independent of the inferred composition of secondary aerosols derived from multiple biogenic VOCs, Atmos. Chem. Phys., 13, 11769–11789, https://doi.org/10.5194/acp-13-11769-2013, 2013.

Berndt, T., Richters, S., Jokinen, T., Hyttinen, N., Kurtén, T., Otkjær, R. V., Kjaergaard, H. G., Stratmann, F., Herrmann, H., Sipilä, M., Kulmala, M., and Ehn, M.: Hydroxyl radical-induced formation of highly oxidized organic compounds, Nat Commun, 7, 13677, https://doi.org/10.1038/ncomms13677, 2016.

Cain, K. P., Liangou, A., Davidson, M. L., and Pandis, S. N.: α-pinene, limonene, and cyclohexene secondary organic aerosol hygroscopicity and oxidation level as a function of volatility, Aerosol Air Qual. Res., 21, 200511, https://doi.org/10.4209/aaqr.2020.08.0511, 2021.

Cheng, Y., Su, H., Koop, T., Mikhailov, E., and Pöschl, U.: Size dependence of phase transitions in aerosol nanoparticles, Nat. Commun., 6, 5923, https://doi.org/10.1038/ncomms6923, 2015.

Couling, S. B., Fletcher, J., Horn, A. B., Newnham, D. A., McPheat, R. A., and Gary Williams, R.: First detection of molecular hydrate complexes in sulfuric acid aerosols, Phys. Chem. Chem. Phys., 5, 4108, https://doi.org/10.1039/B306620G, 2003.

Duplissy, J., DeCarlo, P. F., Dommen, J., Alfarra, M. R., Metzger, A., Barmpadimos, I., Prevot, A. S. H., Weingartner, E., Tritscher, T., Gysel, M., Aiken, A. C., Jimenez, J. L., Canagaratna, M. R., Worsnop, D. R., Collins, D. R., Tomlinson, J., and Baltensperger, U.: Relating hygroscopicity and composition of organic aerosol particulate matter, Atmos. Chem. Phys., 11, 1155–1165, https://doi.org/10.5194/acp-11-1155-2011, 2011.

Dusek, U., Frank, G. P., Curtius, J., Drewnick, F., Schneider, J., Kürten, A., Rose, D., Andreae, M. O., Borrmann, S., and Pöschl, U.: Enhanced organic mass fraction and decreased hygroscopicity of cloud condensation nuclei (CCN) during new particle formation events, Geophys. Res. Lett., 37, 2009GL040930, https://doi.org/10.1029/2009GL040930, 2010.

Engelhart, G. J., Asa-Awuku, A., Nenes, A., and Pandis, S. N.: CCN activity and droplet growth kinetics of fresh and aged monoterpene secondary organic aerosol, Atmos. Chem. Phys., 8, 3937–3949, https://doi.org/10.5194/acp-8-3937-2008, 2008.

Frosch, M., Bilde, M., DeCarlo, P. F., Jurányi, Z., Tritscher, T., Dommen, J., Donahue, N. M., Gysel, M., Weingartner, E., and Baltensperger, U.: Relating cloud condensation nuclei activity and oxidation level of $\alpha$ -pinene secondary organic aerosols: CCN AND OXIDATION LEVEL OF $\alpha$ -PINENE SOA, J. Geophys. Res., 116, n/a-n/a, https://doi.org/10.1029/2011JD016401, 2011.

Fuchs, N. A.: On the stationary charge distribution on aerosol particles in a bipolar ionic atmosphere, Geofis. Pura Appl., 56, 185–193, https://doi.org/10.1007/BF01993343, 1963.

Iyer, S., Rissanen, M. P., Valiev, R., Barua, S., Krechmer, J. E., Thornton, J., Ehn, M., and Kurtén, T.: Molecular mechanism for rapid autoxidation in α-pinene ozonolysis, Nat Commun, 12, 878, https://doi.org/10.1038/s41467-021-21172-w, 2021.

Kang, S., Wildt, J., Pullinen, I., Vereecken, L., Wu, C., Wahner, A., Zorn, S. R., and Mentel, T. F.: Formation of highly oxygenated organic molecules from α-pinene photooxidation: Evidence for the importance of highly oxygenated alkoxy radicals, Atmos. Chem. Phys., 25, 15715–15740, https://doi.org/10.5194/acp-25-15715-2025, 2025.

[revised manuscript text omitted]

---

## Author Comment (AC5)

*Response to Referee #4*

*General Comments*
*The authors investigate the influence mechanisms of sulfuric acid and oxygenated organic molecules on particle growth under varying particle sizes and RH conditions using a custom-built scanning flow condensation particle counter (SFCPC) to measure size-resolved hygroscopicity parameters and organic mass fractions in 3 – 10 nm particles. It further elucidates how the contribution of organic components evolves with particle size, particularly highlighting that increased RH significantly enhances the role of organics in the growth of 5 – 10 nm particles. This finding holds important value for understanding atmospheric new particle formation and growth mechanisms.*

**Responses and Revisions:**

We thank the reviewer for the constructive suggestions and comments concerning our manuscript entitled "Unveiling the organic contribution to the initial particle growth in 3-10 nm size range" [egusphere-2025-4421]. We sincerely appreciate these insightful comments, which have significantly improved both the quality of our manuscript and the methodological rigor of our ongoing research.

We have prepared point-by-point responses to the reviewer's comments below. The original reviewer comments appear in italics. Our responses are presented in blue plain font, while the corresponding revisions in the manuscript appear in blue underlined font.

*1. Comments and suggestions:*
*The authors treat the a-pinene and SO2 as the gas precursors, however, the true components participating particle nucleation should be OOM and H2SO4, which I guess that you did not measure in the current experiment setup. But still, I am wondering that does the [a-pinene]/[SO2] ratio could represent the ambient atmospheric condition?*

**Responses and Revisions:**

We thank the reviewer for raising this important point regarding the atmospheric relevance of our experimental design. We fully agree that atmospheric nucleation is directly driven by the oxidized products (OOMs and $H_2SO_4$). In our experiments, α-pinene and $SO_2$ were used as gas-phase precursors and were oxidized via $O_3$ + $H_2O$ + UV light to generate the components that participating in particle nucleation and subsequent growth.

Regarding the representativeness of the [α-pinene]/[$SO_2$] ratio (0.1-1), we would like to clarify our rationale by combining reported atmospheric field observations and

laboratory experiments. The ambient $SO_2$ concentrations in urban China typically range from 2 to 40 ppbv (Yang et al., 2017; Zhang et al., 2021), while α-pinene levels in forested regions are generally on the order of a few ppbv or below (Li et al., 2023; Magand et al., 2025). However, in laboratory experiments, a wide range of concentration ratios is often employed to systematically investigate relevant processes. For instance, Lehtipalo et al. (2018) explored multi-component new particle formation using ratios spanning over two orders of magnitude (0.5-5 ppbv $SO_2$ and 530-590 ppbv monoterpenes). In our study, the [α-pinene]/[$SO_2$] ratio was set between 0.1 and 1. This range aligns with ratios used in other laboratory studies, such as Xu et al. (2021), who investigated the role of $SO_2$ in biogenic secondary organic aerosol formation with $SO_2$ at 107-112 ppb and γ-terpinene at 102 ppb, and Zhao et al. (2018), who studied the effects of $SO_2$ on α-pinene photooxidation using ~20 ppb α-pinene and ~15 ppb $SO_2$.

Our selection of this concentration ratio was based on both atmospheric observations and previous experimental literature. The data obtained under this ratio—including hygroscopicity, organic fraction, and the dependence on particle size and RH—provide valuable insights that can serve as a reference for understanding atmospheric new particle formation and subsequent growth.

**2. Comments and suggestions:**

*Table 1 and Line 185: The authors claimed that the observed increase in $\kappa$ values at higher RH may be attributed to the production of components with stronger hygroscopicity, it there any direct evidence? What is the product in different RH condition and how to explain their influence on hygroscopicity?*

**Responses and Revisions:**

We thank the reviewer for raising this important point. We acknowledge that our current study did not obtain molecular-level chemical composition data; therefore, we lack direct experimental evidence identifying the specific, more hygroscopic compounds produced at different RH levels in our system.

However, our interpretation is supported by consistent findings from prior literature on analogous α-pinene oxidation systems, which offer both product identification and mechanistic explanations. Yuan et al. (2017) found that at elevated RH, higher $\kappa$ values were observed for secondary organic aerosol (SOA) formed from α-pinene ozonolysis. Through ion chromatography analysis, they directly detected increased levels of formate in the particle phase under humid conditions, linking the elevated $\kappa$ to the presence of specific, more hygroscopic products. Poulain et al. (2010) reported that under humid conditions, the product composition shifted towards compounds containing more oxygenated functional groups, specifically identifying

multifunctional carboxylic acids (e.g., pinonic acid). The formation of such highly oxygenated compounds in the presence of water vapor aligns with the understood chemistry involving stabilized Criegee intermediates. The influence of these products on hygroscopicity could be explained by the fundamental relationship between molecular functionality and $\kappa$. Han et al. (2022) demonstrated that the hygroscopicity of organic compounds varies widely with functional groups and $\kappa$ increased with the functionality in the following order: ($-CH_3$ or $-NH_2$) < ($-OH$) < ($-COOH$ or C=C or C=O). where the carboxylic acid groups are particularly hygroscopic.

Therefore, although we did not directly measure the molecular composition in our experiments, the collective evidence from these studies allows us to reasonably conclude that the observed increase in $\kappa$ at higher RH is likely due to the production of more oxygenated compounds, such as multifunctional carboxylic acids. Their higher intrinsic hygroscopicity, as established by structure-activity relationships, would elevate the overall particle hygroscopicity. We have revised the manuscript text (Section 3.1) to clarify this reasoning and its basis in the literature:

"Contrasting with the negligible size dependence, the measured $\kappa$ values of OOMs exhibited a pronounced increase with RH, rising by 57% from 20% to 80% RH. This finding aligns with Razafindrambinina et al. (2022), who reported higher $\kappa$ values for laboratory-generated α-pinene oxidation products under humid conditions ($\kappa = 0.191$ at 75-80% RH) compared to dry conditions ($\kappa = 0.130$ at RH < 10%). Similarly, Luo et al. (2024) observed that the molecular composition of α-pinene oxidation products evolves with increasing RH. While direct molecular-level speciation from our measurements is unavailable, previous studies on α-pinene oxidation systems provide an explanatory framework. The work of Yuan et al. (2017) suggests that in the presence of water vapor, particles formation may promote the generation of more stable Criegee intermediates, leading to the production of more hygroscopic materials in monoterpene systems. This is supported by evidence of increased formation of oxygenated functional groups, such as multifunctional carboxylic acids, under humid conditions (Poulain et al., 2010). The hygroscopicity of such compounds is intrinsically higher, as the $\kappa$ increases with the functionality in the following order: ($-CH3$ or $-NH2$) < ($-OH$) < ($-COOH$ or C=C or C=O) (Han et al., 2022). Consequently, the observed increase in $\kappa$ values at higher RH in this study is attributed to the likely formation of more hygroscopic components."

**3. Comments and suggestions:**
*Figure 5: It seems that the 3-5 nm particles are more affected by RH compared with larger size, why?*

**Responses and Revisions:**

We apologize for any lack of clarity in our original manuscript that may have led to this interpretation. Upon re-examining our results, we believe the observed trends are not due to the 3-5 nm particles being more affected by RH in an absolute sense, but rather due to the competing influences of the Kelvin effect and the RH-induced changes in the properties of α-pinene oxidation products (increased yields and volatility).

For the larger particles (5-7 nm and 7-10 nm), the Kelvin effect is diminished. Therefore, the primary influence of increased RH is to enhance the contribution of organics by altering the physicochemical properties of the α-pinene oxidation products. This leads to the sharp increase in $f_{org}$ observed at 40% RH, which then stabilizes at higher RH as these effects reach a balance. In contrast, for the smaller 3-5 nm particles, the stronger Kelvin effect presents an energy barrier for condensation of organic vapors. This barrier effectively counteracts the RH-induced enhancement at low to moderate RH. Only when RH increases further dose the driving force for condensation become strong enough to overcome this barrier, leading to the observed increase in $f_{org}$. Thus, the gradual increase in $f_{org}$ for 3-5 nm particles is a manifestation of this delicate balance between two competing mechanisms, highlighting a distinct size-dependent response mechanism to RH, rather than indicating a greater overall sensitivity for the smallest particles.

This interpretation is our inference based on the observed experimental results and supported by the cited literature. To avoid any misunderstanding, we have revised the relevant paragraph in the manuscript (Section 3.3) to articulate this mechanism more clearly:

"We speculate that these behaviours arise from the competing influences of humidity on the physicochemical properties of α-pinene oxidation products and the Kelvin effect. For such small nanoparticles, the partitioning of a molecule into the particulate phase is influenced by both its volatility and the Kelvin effect (Riipinen et al., 2012). Previous molecular measurements in both gas and particle phases have reported increased yields at elevated RH (Poulain et al., 2010). Concurrently, Surdu et al. (2023) observed that α-pinene oxidation products become more volatile under humid conditions. The relative stability of $f_{org}$ in the 3-5 nm particles at low RH condition may thus be explained by a balance between these two competing mechanisms, where the heightened Kelvin effect presents a significant barrier to condensation. For larger particles, the diminished Kelvin effect facilitates the condensation of organic compounds, allowing even more volatile products to contribute to nanoparticle growth. The distinct response patterns, where the enhancement occurred gradually for 3-5 nm

particles but sharply for larger particles, suggest that the Kelvin effect plays a more dominant role for the smallest particle growth at lower RH."

**Refences:**

Han, S., Hong, J., Luo, Q., Xu, H., Tan, H., Wang, Q., Tao, J., Zhou, Y., Peng, L., He, Y., Shi, J., Ma, N., Cheng, Y., and Su, H.: Hygroscopicity of organic compounds as a function of organic functionality, water solubility, molecular weight, and oxidation level, Atmos. Chem. Phys., 22, 3985–4004, https://doi.org/10.5194/acp-22-3985-2022, 2022.

Lehtipalo, K., Yan, C., Dada, L., Bianchi, F., Xiao, M., Wagner, R., Stolzenburg, D., Ahonen, L. R., Amorim, A., Baccarini, A., Bauer, P. S., Baumgartner, B., Bergen, A., Bernhammer, A.-K., Breitenlechner, M., Brilke, S., Buchholz, A., Mazon, S. B., Chen, D., Chen, X., Dias, A., Dommen, J., Draper, D. C., Duplissy, J., Ehn, M., Finkenzeller, H., Fischer, L., Frege, C., Fuchs, C., Garmash, O., Gordon, H., Hakala, J., He, X., Heikkinen, L., Heinritzi, M., Helm, J. C., Hofbauer, V., Hoyle, C. R., Jokinen, T., Kangasluoma, J., Kerminen, V.-M., Kim, C., Kirkby, J., Kontkanen, J., Kürten, A., Lawler, M. J., Mai, H., Mathot, S., Mauldin, R. L., Molteni, U., Nichman, L., Nie, W., Nieminen, T., Ojdanic, A., Onnela, A., Passananti, M., Petäjä, T., Piel, F., Pospisilova, V., Quéléver, L. L. J., Rissanen, M. P., Rose, C., Sarnela, N., Schallhart, S., Schuchmann, S., Sengupta, K., Simon, M., Sipilä, M., Tauber, C., Tomé, A., Tröstl, J., Väisänen, O., Vogel, A. L., Volkamer, R., Wagner, A. C., Wang, M., Weitz, L., Wimmer, D., Ye, P., Ylisirniö, A., Zha, Q., Carslaw, K. S., Curtius, J., Donahue, N. M., Flagan, R. C., Hansel, A., Riipinen, I., Virtanen, A., Winkler, P. M., Baltensperger, U., Kulmala, M., and Worsnop, D. R.: Multicomponent new particle formation from sulfuric acid, ammonia, and biogenic vapors, Sci. Adv., 4, eaau5363, https://doi.org/10.1126/sciadv.aau5363, 2018.

Li, Q., Han, Y., Huang, D., Zhou, J., Che, H., Zhang, L., Lu, K., Yang, F., and Chen, Y.: Springtime reactive volatile organic compounds (VOCs) and impacts on ozone in urban areas of yunnan-guizhou plateau, China: A PTR-TOF-MS study, Atmos. Environ., 307, 119800, https://doi.org/10.1016/j.atmosenv.2023.119800, 2023.

Luo, H., Guo, Y., Shen, H., Huang, D. D., Zhang, Y., and Zhao, D.: Effect of relative humidity on the molecular composition of secondary organic aerosols from α-pinene ozonolysis, Environ. Sci.: Atmos., 4, 519–530, https://doi.org/10.1039/D3EA00149K, 2024.

Magand, O., Boulanger, P., Staménoff, P., David, M., Hernandez, P., Golubic, E., Hello, Y., Ah-Peng, C., Duflot, V., Ktata, O., and Rocco, M.: Monitoring and volatile organic compounds characterization (isoprene, monoterpene and BTEX) in a tropical-oceanic environment in reunion island (Indian ocean, south hemisphere), Front. Environ. Sci., 13, https://doi.org/10.3389/fenvs.2025.1704158, 2025.

Poulain, L., Wu, Z., Petters, M. D., Wex, H., Hallbauer, E., Wehner, B., Massling, A., Kreidenweis, S. M., and Stratmann, F.: Towards closing the gap between hygroscopic growth and CCN activation for secondary organic aerosols – Part 3: Influence of the

chemical composition on the hygroscopic properties and volatile fractions of aerosols, Atmos. Chem. Phys., 10, 3775–3785, https://doi.org/10.5194/acp-10-3775-2010, 2010.

Razafindrambinina, P. N., Malek, K. A., Dawson, J. N., DiMonte, K., Raymond, T. M., Dutcher, D. D., Freedman, M. A., and Asa-Awuku, A.: Hygroscopicity of internally mixed ammonium sulfate and secondary organic aerosol particles formed at low and high relative humidity, Environ. Sci.: Atmos., 2, 202–214, https://doi.org/10.1039/D1EA00069A, 2022.

Riipinen, I., Yli-Juuti, T., Pierce, J. R., Petäjä, T., Worsnop, D. R., Kulmala, M., and Donahue, N. M.: The contribution of organics to atmospheric nanoparticle growth, Nat. Geosci., 5, 453–458, https://doi.org/10.1038/ngeo1499, 2012.

Surdu, M., Lamkaddam, H., Wang, D. S., Bell, D. M., Xiao, M., Lee, C. P., Li, D., Caudillo, L., Marie, G., Scholz, W., Wang, M., Lopez, B., Piedehierro, A. A., Ataei, F., Baalbaki, R., Bertozzi, B., Bogert, P., Brasseur, Z., Dada, L., Duplissy, J., Finkenzeller, H., He, X.-C., Höhler, K., Korhonen, K., Krechmer, J. E., Lehtipalo, K., Mahfouz, N. G. A., Manninen, H. E., Marten, R., Massabò, D., Mauldin, R., Petäjä, T., Pfeifer, J., Philippov, M., Rörup, B., Simon, M., Shen, J., Umo, N. S., Vogel, F., Weber, S. K., Zauner-Wieczorek, M., Volkamer, R., Saathoff, H., Möhler, O., Kirkby, J., Worsnop, D. R., Kulmala, M., Stratmann, F., Hansel, A., Curtius, J., Welti, A., Riva, M., Donahue, N. M., Baltensperger, U., and El Haddad, I.: Molecular understanding of the enhancement in organic aerosol mass at high relative humidity, Environ. Sci. Technol., 57, 2297–2309, https://doi.org/10.1021/acs.est.2c04587, 2023.

Xu, L., Tsona, N. T., and Du, L.: Relative Humidity Changes the Role of SO2 in Biogenic Secondary Organic Aerosol Formation, J. Phys. Chem. Lett., 12, 7365–7372, https://doi.org/10.1021/acs.jpclett.1c01550, 2021.

Yang, X., Wang, S., Zhang, W., Zhan, D., and Li, J.: The impact of anthropogenic emissions and meteorological conditions on the spatial variation of ambient SO2 concentrations: A panel study of 113 Chinese cities, Science of The Total Environment, 584–585, 318–328, https://doi.org/10.1016/j.scitotenv.2016.12.145, 2017.

Yuan, C., Ma, Y., Diao, Y., Yao, L., Zhou, Y., Wang, X., and Zheng, J.: CCN activity of secondary aerosols from terpene ozonolysis under atmospheric relevant conditions, J. Geophys. Res.-Atmos., 122, 4654–4669, https://doi.org/10.1002/2016JD026039, 2017.

Zhang, X., Wang, Z., Cheng, M., Wu, X., Zhan, N., and Xu, J.: Long-term ambient SO2 concentration and its exposure risk across China inferred from OMI observations from 2005 to 2018, Atmos. Res., 247, 105150, https://doi.org/10.1016/j.atmosres.2020.105150, 2021.

Zhao, D., Schmitt, S. H., Wang, M., Acir, I.-H., Tillmann, R., Tan, Z., Novelli, A., Fuchs, H., Pullinen, I., Wegener, R., Rohrer, F., Wildt, J., Kiendler-Scharr, A., Wahner, A., and Mentel, T. F.: Effects of NO$_x$ and SO$_2$ on the secondary organic aerosol formation from

photooxidation of α-pinene and limonene, Atmos. Chem. Phys., 18, 1611–1628, https://doi.org/10.5194/acp-18-1611-2018, 2018.